METEORv1.0.1: A novel framework for emulating multi-timescale regional climate responses

Marit Sandstad<sup>1</sup>, Norman Julius Steinert<sup>1</sup>, Susanne Baur<sup>2,3</sup>, and Benjamin Mark Sanderson<sup>1</sup>

<sup>1</sup>CICERO Center for International Climate Research, Oslo 0349, Norway

<sup>2</sup>CECI, Université de Toulouse, CERFACS, CNRS, Toulouse, France

<sup>3</sup>CNRM, Université de Toulouse, Météo-France/CNRS, Toulouse, France

**Correspondence:** Marit Sandstad (marit.sandstad@cicero.oslo.no)

Abstract. Resolved spatial information for climate change projections is critical to any robust assessment of climate impacts and adaptation options. However, the range of spatially resolved future scenario assessments available is limited, due to the significant computational and human demands of Earth System Model (ESM) pipelines. In order to explore a wider variety of societal outcomes and to enable coupling of climate impacts into societal modelling frameworks, rapid spatial emulation of ESM responses to climate change is therefore desirable. Many existing pattern scaling methods assume spatial climate signals which scale linearly with global temperature change, where the pattern of response is independent of the nature and timing of emissions. However, this assumption may introduce biases in emulated climates, especially under net negative emissions and overshoot scenarios. To address these biases, we propose a novel emulation system, METEOR, which represents multitimescale spatial climate responses to multiple climate forcers. The mapping of emissions to forcing is provided by the CICERO Simple Climate Model, combined with a calibration system that can be used to train model-specific pattern response engines using only core training simulations from CMIP. Here, we demonstrate that our fitted spatial emulation system is capable of rapidly and accurately predicting gridded annual mean temperature and precipitation responses to out-of-sample scenarios.

Copyright statement. TEXT

### 1 Introduction

5 Spatially resolved information is essential for informing robust assessments of mitigation and adaptation strategies in response to global climate change (IPCC, 2021, 2022). Accurate and detailed regional projections enable policymakers and stakeholders to understand potential impacts and to plan accordingly (IPCC, 2022). The Coupled Model Intercomparison Project (CMIP) aims to deliver this information, and is increasingly moving towards an operational procedure, wherein CMIP7 Earth System Models will be run on a semi-regular frequency, allowing updates of model complexity, historical forcing and future scenarios (Dunne et al., 2024). However, these pipelines are time-consuming and computationally intensive, resolving numerous physical, chemical, and biological processes at high spatial and temporal resolutions such that a modest number of scenario simulations with multiple models takes years to achieve (Eyring et al., 2016). Demand for regional climate information increas-

ingly requires more frequent updates for a wider range of policy-relevant future scenarios. Limitations on time, computational and human resources needed for ESM simulations constrain the range of future scenarios and models that can be feasibly explored (Nicholls et al., 2020, 2021). In addition, there is an increased need for fast spatial modelling frameworks where regional climate impacts are resolved to allow for the simulation of risks, inequalities, impacts on and interactions between social, economic and natural systems under climate change (van Vuuren et al., 2012; Kikstra et al., 2021; Dietz et al., 2021; Ferrari et al., 2022; James Rising, 2022).

To address these needs, spatial emulation techniques have been developed to rapidly approximate the output of ESMs under various scenarios (Zelazowski et al., 2018). Linear pattern scaling is one such approach, which assumes that spatial patterns of climate response scale linearly with global mean temperature change (Santer et al., 1990; Mitchell, 2003; Beusch et al., 2020). This method allows for quick estimations of regional climate change by scaling predefined spatial patterns according to projected global temperature changes (Tebaldi et al., 2021; Zhao et al., 2017).

Several models have exploited pattern scaling techniques in climate research. The SCENGEN tool, for instance, generates regional climate change scenarios by scaling standardised patterns derived from General Circulation Models (GCMs) (Hulme et al., 2000). Similarly, the MESMER framework employs pattern scaling to efficiently emulate temperature and precipitation fields from ESMs (Beusch et al., 2020, 2022). Likewise, the PRIME framework makes use of pattern scaling and subsequently provides the spatial climate information as input for a land surface model (Mathison et al., 2024), allowing more direct simulation of terrestrial ecosystems and downstream human and economic impacts. The STITCHES model uses a different approach, by splicing together portions of existing simulations with global mean temperature and its derivative corresponding to the desired prediction to emulate scenarios which have not yet been simulated (Tebaldi et al., 2022). This allows the model to represent, for example, differences between resolved patterns under warming and cooling climate states, but is limited by the finite number of climate analogues in the training dataset - which in CMIP is relatively sparse, and requires concatenation of segments of simulations which may produce unphysical discontinuities in the emulated climate. Further, the ESEm framework (Watson-Parris et al., 2021) provides options to build various types of more process agnostic machine learning based emulators for spatio-temporally resolved data, using approaches such as Random Forest, Gaussian Process and Neural Network regressions. Comparison of these methods to traditional linear pattern scaling shows good performance (Watson-Parris et al., 2022), but the setup used requires output from experiments which have only been run by a smaller subset of the CMIP6 ESMs. Similarly, Mansfield et al. (2020) utilised the specific-per-species modelling intercomparison setup and output in PDRMIP (Myhre et al., 2017), to define short- and long-term ESM responses using Gaussian Process regression, which can then in turn be used to emulate long-term responses to new scenarios from short-term responses. Due to the data requirements for training to new models, this setup, though potentially powerful, is not immediately applicable to CMIP6 or new emerging ESM datasets.

While each of these methods have increased capacity to produce regional climate projections with reduced computational demands, questions remain on how to emulate hysteresis, forcing dependency and nonlinear responses in future climate, which have been demonstrated to exist in Earth System Models (Sanderson et al., 2024) but can be precluded by emulator assumptions. Pattern scaling models such as MESMER rely on the assumption that spatial patterns of climate response are a singular function of global mean temperature, insensitive to the forcing history and the emission trajectory (Collins et al., 2013; Zhao et al.,

2017). Patterns of warming in PRIME are also subject to this linearity assumption, though its process-based land surface model component could potentially represent memory in slow-timescale terrestrial processes if feedbacks were included. STITCHES can potentially resolve non-linear and time-emergent behaviour to the degree that behaviour is represented in the training scenarios, but is limited to the degree it can generalise hysteresis and climate reversibility dynamics. This linearity assumption allows reasonable performance under scenarios of gradual and monotonic climate change but can introduce significant biases under strong mitigation scenarios or scenarios involving overshoots in greenhouse gas concentrations (Herger et al., 2015; Good et al., 2015; Tebaldi and Knutti, 2007). Additionally, non-linearities in the climate system, such as feedback mechanisms and varying climate sensitivities over time (Jonko et al., 2013), can lead to time-evolving and forcing-dependent spatial patterns that are not adequately captured by traditional pattern scaling approaches (Huntingford et al., 2000; Shiogama et al., 2010).

To address these limitations, we propose METEOR (Multivariate Emulation of Time-Evolving and Overlapping Responses), a novel emulation framework that accounts for spatial climate responses to a range of climate forcers emerging over different timescales by using impulse response assumptions applied to a spatially resolved basis set. Figure 1 illustrates how METEOR can produce non-linear and hysteresis responses to an idealised forcing trajectory. METEOR builds on the emissions-forcing engine from the CICERO Simple Climate Model (C-SCM; Sandstad et al., 2024), and can represent the temporal evolution and forcing dependency of spatial patterns, providing regional climate projections which preserve the hysteresis and time-evolving response to forcing present in the target Earth System Model, allowing more consistent representation of scenarios involving overshoots or pathways with diverse mixes of short- and long-lived climate forcers.

In this paper, we present the structure and validation of METEOR. We demonstrate how this approach can capture nonlinear and time-dependent aspects of regional climate change. As such, METEOR offers a practical tool for researchers to rapidly explore a wide variety of societal outcomes and to assess mitigation and adaptation options informing policymakers with greater confidence.

### 2 Methods

The methodology for METEOR is illustrated in Figure 2, and described below. The framework of METEOR is based on the assumption that a step change in a given climate forcer can induce a number of time-evolving patterns for a predicted output variable, each of which emerges on a specific timescale. The METEOR framework allows groups of climate forcers to be associated with a number of time-evolving pattern responses, which in the current version then combine linearly to give the total climate response.

The primary forcer component of climate change is the greenhouse gas (GHG) signal due to well-mixed greenhouse gases, which exert a forcing on the climate system (Forster et al., 2025). For METEOR, we assume that the time-evolving pattern of response to GHG forcing can be approximated by the response of the climate system to a step change in CO<sub>2</sub> concentrations. This is practical given the ready availability of *abrupt4x-CO*<sub>2</sub> simulations (in which atmospheric carbon dioxide levels are instantaneously quadrupled from pre-industrial levels and the system is allowed to evolve for at least 140 years) for all Earth

**Figure 1. METEOR pattern scaling hysteresis.** Illustration of the emulated METEOR ensemble response showing a) global mean temperature (GMST) and b) precipitation (GMP) response to an idealised radiative forcing with a Gaussian ramp-up to 1 W/m<sup>2</sup> and subsequent ramp-down within 500 years. c) shows the emulated relationship between GMST and GMP, compared with common pattern scaling, which various patterns scaling approaches are based on. METEOR (grey), trained by CMIP6 models (see Methods), is able to emulate hysteresis behaviour, compared to a common pattern scaling (blue) that uses linear regression of the relationship between GMST and GMP. Each individual grey line shows the response of METEOR trained to emulate a specific CMIP6 model.

System Models in the CMIP archive, and was found to be a reasonable approximation in prior process modelling studies which considered the pulse-response to a number of different greenhouse gases (Myhre et al., 2017).

METEOR assumes that the response to a step change in forcing from a given source can be represented by the sum of one or more impulse response patterns, each with its own timescale of emergence as represented by a decaying exponential timeseries for pattern saturation. Each timescale and corresponding pattern can capture elements of the physical response which emerge at different timescales, such that different spatial patterns can be associated, for example, with the warming of the shallow and deep ocean. Similarly, some forcing agents such as sulfate aerosols and black carbon are associated with markedly different warming patterns and timescales to those of well-mixed greenhouse gases (Myhre et al., 2017). The METEOR framework allows for individual forcers, or groups of forcers, to be associated with their own set of time-emergent patterns, allowing for the model to simulate and distinguish between the spatial pattern of climate change associated with different forcer types. As such, METEOR can be trained employing only steps 1-4 (Fig. 2), for a single forcer version, or steps 1-4 can be repeated for multiple forcers for which abrupt step-change forcing experiment data are available.

In practice, METEOR uses outputs of the *abrupt4x-CO*<sub>2</sub> experiment to fit the GHG response. Then, as separate step-change experiment data is not generally available in CMIP for other forcers, a residual signal from a *historical* and single future scenario run (O'Neill et al., 2016) is used for inverse estimation of patterns and timescales for a sulfate aerosol response.

We choose this technique to distinguish sulfate aerosol only, as this is the most different and impactful non-GHG forcer (Myhre et al., 2017; Forster et al., 2025). See also Figure 7.7 of (Forster et al., 2021) where the aerosol forcing is currently the most uncertain and largest non-GHG contribution (though, this may change in the future (Adams et al., 2001; Liao and Seinfeld, 2005; Bauer et al., 2007; Liao et al., 2009; Bellouin et al., 2011; Hauglustaine et al., 2014), which is a source of structural uncertainty in the setup of METEOR demonstrated here). In the setup we use here, the totality of the aerosol-cloud-interactions are represented as a function of sulfate aerosol forcing, and a substantial part of the aerosol radiation forcing comes from sulfate forcing. All other forcer responses including tropospheric and stratospheric ozone, BC- and OC- aerosol direct forcing, and stratospheric water vapour forcing are mapped from forcing strength using the GHG-response patterns and timescales. A further breakdown of forcer types requires additional dedicated experiments. For convenience, we will denote this part of the pattern simply as aerosol patterns, although it is both more specific (only fitted for sulfate aerosol) and less so (as it is a residual pattern, and will naturally also pick up other non-sulfate or even non-aerosol patterns). The obtained responses and patterns can then be used to emulate the spatial response to a previously unseen experiment for which there is emissions (or concentrations) data available.

As emissions or concentration inputs, rather than forcing inputs, are available for both training and emulated scenarios, METEOR uses the emissions-forcing engine from C-SCM (Sandstad et al., 2024) to map from emissions or concentration inputs to forcing signals. For each modelled forcer type, the forcing input signal is scaled by C-SCM's own forcing strength in the training scenario.

This METEOR code framework is available at https://github.com/benmsanderson/METEOR and is importable as a Python library. The methodology does not come pre-trained, but includes tools to read local model output training data, and supporting functionality to download data on appropriate formats. In this article, we present results for a selection of CMIP6 models applied to yearly temperature and precipitation, but the methodology is not limited to these variables, or to the set of ESM model output used here. Indeed any yearly variable output can in principle be emulated. Though, the reliability of the fit must be assessed by the user. The computational time for training is relatively fast (order of minutes on a laptop), but performance depends on the resolution of the ESM model target for emulation, and local machine specifications including memory limitations. Overall the code allows for efficient emulation of models in the CMIP6 archive and computation and application to a variety of climate scenarios.

# 2.1 Training the model: Construction of transient spatial response patterns

In this section we describe how we use the  $abrupt4xCO_2$  experiment to find impulse response timescales and patterns for  $CO_2$  that can be used to recreate and model GHG forcing response in various experiments. A combination of historical and scenario data is then employed to estimate separate timescales and patterns for sulfate aerosol forcing. These estimates are then added linearly to the GHG forcing and patterns to provide the composite response.

**Figure 2. METEOR flowchart.** Illustration of the concept and sequence of METEOR and its integration with the CICERO simple climate model. Note that the side panels represent the necessary preparatory steps for the creation of GHG (left) and aerosol (right) transient spatial climate response pattern. Here, the aerosol pattern calculations, specifically step 7, requires input from the GHG pattern calculation steps 3 and 4 (dotted gray line). Hence, steps 1–10 are required to train METEOR (centre panels; also see section 2.1). Their outcome is used in step C in METEOR (see section 2.2). Note that METEOR partly integrates the C-SCM as it makes use of its emissions-to-forcings module.

### 2.1.1 Greenhouse gas response estimation

We begin by obtaining the annual mean outputs for the target ESM we wish to emulate. The variables of interest in this study are surface air temperature (tas) and precipitation (pr). To obtain a greenhouse gas response signal, we use the abrupt4x- $CO_2$  experiment, in which atmospheric  $CO_2$  concentrations are instantaneously quadrupled relative to pre-industrial levels, and the piControl (pre-industrial control) simulation (Eyring et al., 2016, Fig. 2, step 1). We obtain the radiative forcing time series for GHGs from the abrupt4x- $CO_2$  emissions by using the C-SCM emission-to-forcing module (Sandstad et al., 2024) that converts carbon emissions into climate forcing (Fig. 2, step 2).

For the given target variable, we estimate the gridded climate response to increased CO<sub>2</sub> by calculating the anomaly, subtracting the *piControl* climatology (full time-series mean) from the *abrupt4xCO*<sub>2</sub> simulation to yield **X**, a matrix of dimensions  $s \times t$ , with s being the number of spatial grid points (of the input ESM resolution as METEOR works on the input data resolution with no regridding included) and t the number of years in the simulation with the values for the variable of interest in each point in space and time as its data. From  $\mathbf{X}_{s,t}$  we calculate the global mean anomaly time series  $\mathbf{x}_{global}(t)$  by performing an area-weighted average (weighting by  $\cos(\operatorname{lat})$ ) over all spatial grid points:

$$\mathbf{x}_{\text{global}}(t) = \frac{1}{s} \sum_{i=1}^{s} w_i \cdot \mathbf{X}_{i,t},\tag{1}$$

where  $w_i$  is the area weight for pixel i, with a total number of pixels s.

We then find an approximate representation of this global mean response as a sum of n exponential decay functions, representing different climate response timescales:

$$\mathbf{x}_{\text{global}}(t) \approx \sum_{k=1}^{n} a_k \left( 1 - e^{-t/\tau_k} \right),\tag{2}$$

where  $a_k$  are amplitudes and  $\tau_k$  are the decay timescales to be determined for mode k. Note that (Womack et al., 2025) presents a related approach to emulate from forcing to temperature using a Green-functions mapping which includes per grid-point impulse response functions, rather than METEOR's global impulse response functions.

METEORv1.0 can decompose the global mean response into a user-defined number of timescales, with each added timescale fitted in an exponentially longer and non-overlapping time range so that  $\tau_k \in (10^k, 10^{k+1})$ , minimised from an initial guess of  $\tau_k^{\rm guess} = 5 \cdot 10^k$ . This allows for a structured separation of the timescales. Three timescales (n=3), which will be referred to as inter-annual mode (1-10 years), inter-decadal mode (10-100 years) and inter-centennial mode (100-1000 years) response, is the configuration we will describe here. Note that choice of number of timescales (n) can be practically informed by assessing the point at which there is no further improvement in performance in the approximation detailed in Eq. 2 (see section 3 for an illustration).

For the timescale decomposition, we construct a matrix  $T_{GHG}$  of dimensions  $n \times n_t$ , where each row n corresponds to an exponential decay function with a specific timescale, and  $n_t$  refers to the number of simulation years (Fig. 2, step 3). Trepresents the temporal evolution of the global mean climate response across different timescales:

$$\mathbf{T}_{\text{GHG}k,t} = 1 - e^{-t/\tau_k}, \quad \text{for } k = 1, 2, \dots, n.$$
 (3)

Assuming that the spatiotemporal response can be represented as a sum of patterns which each emerge as a saturating exponential decay (Proistosescu and Huybers, 2017), we express the anomaly matrix  $\mathbf{X}$  as a product of spatial patterns and temporal basis functions, where  $\mathbf{B}$  is the spatial pattern matrix of dimensions  $s \times n$ :

$$\mathbf{X} = \mathbf{B}_{\text{GHG}} \cdot \mathbf{T}_{\text{GHG}}.\tag{4}$$

This assumption allows the calculation of the spatial response matrix  $\mathbf{B}$  (Fig. 2, step 4), given that we have a prior estimate of the timescale matrix  $\mathbf{T}$  and the full time-evolving output from the target model,  $\mathbf{X}$ . Hence, we solve for  $\mathbf{B}$  with a least-squares estimate  $\mathbf{T}^+$  (Moore-Penrose Pseudoinverse) (Barata and Hussein, 2012) of the matrix  $\mathbf{T}$ . Consequently, the derived values of  $\mathbf{B}$  minimise the residuals between the observed and reconstructed anomalies:

$$\mathbf{B}_{\mathrm{GHG}} = \mathbf{X} \cdot \mathbf{T}_{\mathrm{GHG}}^{+}. \tag{5}$$

Finally, we normalise the spatial patterns by the effective radiative forcing  $F_{4\times}$  associated with the quadrupling of  $CO_2$  concentrations in the C-SCM, yielding a spatial pulse-response function  $\widetilde{\mathbf{B}}_{GHG} = \frac{\mathbf{B}_{GHG}}{F_{4\times CO_2}}$  that represents the spatial climate response pattern per unit forcing.

Estimating the climate response of any forcer species for METEOR in the way described here for CO<sub>2</sub> is possible, but direct estimation requires a species-specific forcing step-change experiment equivalent to the *abrupt4xCO*<sub>2</sub>. Such experiments have only been performed for a limited number of CMIP5 generation of models (Myhre et al., 2017)

### 2.1.2 Aerosol response estimation

To construct the aerosol response patterns in the CMIP6 emulation, we instead make use of transient experiments to inversely calculate the aerosol spatial pulse response functions, constructing a residual between the ESM output for an all-forcing ( $F_{ALL}$ ) experiment (where both greenhouse gases and aerosols are varying) and a synthetic GHG forcing ( $F_{GHG}$ ) experiment, the response for which we estimate using the METEOR GHG pattern response above. This assumes that such a residual is primarily explained by the aerosol climate response. In Section 3 we verify that this assumption yields reasonable results in the scenarios considered for CMIP6.

Similarly to how we found X for the  $abrupt4xCO_2$ , we obtain the full ESM scenario response  $S_{s,t'}$  (dimensions  $s \times n'$ , where n' is now the number of years in the scenario) from the CMIP model output corresponding to the emission scenario which will be used in the training process. For the all-forcing scenario, we utilise a combination of the historical experiment and the Shared Socioeconomic Pathway SSP2-4.5 (Eyring et al., 2016) from CMIP6 ESM output (Fig. 2, step 5), though any

experiment including all forcers could in principle be used. Here again, we obtain the aerosol forcing time series from the *historical* and *SSP2-4.5* emissions by using the C-SCM emission-to-forcing module (Fig. 2, step 6).

Further, let  $F_{\rm GHG}(t')$  and  $F_{\rm aer}(t')$  denote the GHG and aerosol forcings at time  $t'=1,2,\ldots,n$ , respectively. From the SSP2-4.5 scenario, we compute the incremental changes in GHG forcing  $\Delta F_{\rm GHG}(t')=F_{\rm GHG}(t')-F_{\rm GHG}(t'-1)$ , with  $\Delta F_{\rm GHG}(1)=F_{\rm GHG}(1)$ . We can then calculate the time-evolving coefficients by convolving the incremental GHG forcings with the exponential decay functions:

200 
$$\mathbf{C}_{GHG_{i,t}} = \sum_{t'=1}^{t} \Delta F_{GHG}(t') \cdot (1 - e^{-(t-t')/\tau_i}),$$
 (6)

or in matrix notation:

$$\mathbf{C}_{GHG} = \{ \Delta \mathbf{F}_{GHG} * \mathbf{T}_{GHG} \}(t), \tag{7}$$

where {} indicates a convolution over the time dimension. From that, the estimated GHG-induced spatial response is then:

$$\mathbf{S}_{\mathrm{GHG}_{s,t'}} = \sum_{i=1}^{N} \widetilde{\mathbf{B}}_{\mathrm{GHG}_{s,i}} \cdot \mathbf{C}_{\mathrm{GHG}_{i,t'}}.$$
(8)

Using the total scenario response  $S_{s,t'}$  from the ESM simulations of SSP2-4.5, the aerosol-induced response can be estimated from the residual of subtracting the GHG response from the total response (Fig. 2, step 7):

$$\mathbf{S}_{\text{resid}_{s,t'}} = \mathbf{S}_{s,t'} - \mathbf{S}_{\text{GHG}_{s,t'}}.\tag{9}$$

As was done for the GHG response, we here assume the aerosol response can be represented with  $n_{\text{aer}} = 3$  timescales  $\tau_{\text{aer}}$  for the inter-annual, inter-decadal and inter-centennial responses, respectively. The time response matrix  $\mathbf{T}_{\text{aer}}$  for each timescale j for a step change in aerosol forcing is constructed as (Fig. 2, step 8):

$$T_{aer_{j,t'}} = 1 - e^{-t'/\tau_{aer_{j}}}.$$
 (10)

The time-evolving coefficients for the aerosol pattern response in the scenario can then be calculated by convolving the aerosol forcing difference time series with the synthetic pulse response time series:

$$C_{aerj,t} = \sum_{t'=1}^{t} \Delta F_{aer}(t') T_{aerj,t-t'}, \tag{11}$$

or in matrix notation:

$$\mathbf{C}_{\text{aer}} = \{ \Delta \mathbf{F}_{\text{aer}} * \mathbf{T}_{\text{aer}} \} (t). \tag{12}$$

In order to find optimal values for  $\tau_{aer}$  (introduced in Section 2.1.2), we use an optimization algorithm to search for values of  $\tau_{aer}$  which minimise the error in the projection of the global mean target field residual timeseries (e.g. temperature, precipitation) onto the basis defined by  $\mathbf{C}_{aer}$  (Fig. 2, step 9). Once  $\mathbf{C}_{aer}$  is known, we can create a least-squares estimate of the spatial patterns of aerosol response  $\mathbf{B}_{aer}$  as the product of the residual matrix with the Moore-Penrose Pseudoinverse (Barata and Hussein, 2012)  $\mathbf{C}_{aer}^+$  of the coefficients matrix  $\mathbf{C}_{aer}$ :

$$\mathbf{B}_{\text{aer}} = \mathbf{S}_{\text{resid}} \cdot \mathbf{C}_{\text{aer}}^{+}. \tag{13}$$

The emulated spatial aerosol response  $S_{aer}$  for a novel forcing timeseries  $\Delta F$  can then be computed by convolving with the timescale response matrix  $T_{aer}$  and taking the dot product with the aerosol spatial response patterns  $B_{aer}$  (Fig. 2, step 10):

$$\mathbf{S}_{\text{aer}} = \mathbf{B}_{\text{aer}} \cdot \mathbf{C}_{\text{aer}} = \mathbf{B}_{\text{aer}} \cdot \{ \mathbf{\Delta} \mathbf{F} * \mathbf{T}_{\text{aer}} \} (t)$$
(14)

## 2.2 Applying the model: Multi-forcer multi-timescale pattern scaling in out-of-sample scenarios

The calculations of the transient spatial climate response patterns for GHG and aerosol forcing outlined in the previous sections can now be used to emulate the spatio-temporal climate response of any climate scenario. For the application in a new (out-of-sample) emissions scenario, METEOR converts any given emission scenario (Fig. 2, step A) into forcing time series using the emissions-to-forcings module of the C-SCM (Fig. 2, step B). Convolving these time series with the GHG and aerosol patterns (Fig. 2, step C) can then be utilised to reconstruct the total multi-forcer multi-timescale emulated climate response by linearly combining the GHG and aerosol responses (Fig. 2, step D):

$$\mathbf{S}_{\text{emul}_{s,t'}} = \mathbf{S}_{\text{GHG}_{s,t'}} + \mathbf{S}_{\text{aer}_{s,t'}},\tag{15}$$

where  $\mathbf{S}_{GHGs,t'}$  and  $\mathbf{S}_{aers,t'}$  are obtained according to equations (6, 8) and (11, 14) respectively, with forcing timeseries  $\Delta F_{GHG}(t')$  and  $\Delta F_{aer}(t')$  calculated for the emissions scenario of interest.

#### 3 METEOR evaluation


To evaluate the performance of METEOR, we have trained the model on CMIP6 data for a large number of CMIP6 models (see Tab. B1 for a full list) using a training dataset that consists of *abrupt4x-CO*<sub>2</sub>, *piControl*, *historical* and *SSP2-4.5* modelling output in each case. Comparing to CMIP6 output from these experiments, we show the in-sample accuracy. Furthermore, we

have applied the resulting emulation models to a number of additional scenarios from ScenarioMIP (Tebaldi et al., 2021): SSP1-2.6, SSP3-7.0, SSP5-8.5 and SSP5-3.4-over to consider the out-of-sample performance (note that SSP5-3.4-over was only performed by a limited number of models).

Additionally, in Appendix A we explore the effect and performance of the model for the *abrupt4x-CO*<sub>2</sub> experiment and the *historical* and SSP2-4.5 experiment combination depending on the number of timescales used, showing that three timescales seem to yield good performance. Appendix B shows per model results including Table B1 which lists the values for the timescales obtained for GHG and aerosol response for Surface Air Temperature (tas) and Precipitation (pr) for each model and thus also serves as a reference of which models were included in the analysis.

# 3.1 GHG and aerosol multi-timescale pattern of climate change






Starting with the GHG response, Figure 3 shows the full emulation and contributions from the different timescale patterns obtained from the *abrupt4x-CO*<sub>2</sub> experiment for temperature and precipitation. Panels a and f show the CMIP6 multi-model mean outputs averaged over years 80–120 for the experiment. Panels b and g show the multi-model mean average of the METEOR emulations for the same models and time period. Below in panels c and h, d and i and e and j, of Figure 3 decomposes the model mean emulation into components associated with the inter-annual, inter-decadal and inter-centennial responses for temperature (c, d and e) and precipitation (h, i and j). The patterns show Arctic amplification (temperature) in both inter-annual and inter-decadal modes. In the inter-decadal temperature mode (d) METEOR picks up on a North-Atlantic relative cooling anomaly, which is induced by the nonlinear response of the Atlantic Meridional Overturning Circulation in some CMIP6 models. The fast mode response for precipitation is dominated by diverse responses over the tropical Pacific, with more widespread changes in the slower modes.

Figure 4 similarly shows multi-model mean CMIP6 data (panels a and g) and emulation (panels b and h), and the split into contributions from the various timescales and patterns. Here the average has been taken over years 1980–2020 in the combined *historical* and *SSP2-4.5* experiment. Panels c and i show the combined GHG responses for temperature and precipitation, whereas panels d and j, e and k, and f and l show the inter-annual, inter-decadal and inter-centennial aerosol contributions. Broadly speaking, the inter-annual aerosol timescale provides an overall cooling or drying effect. The inter-decadal aerosol signal dampens the Arctic amplification and counteracts the spatial pattern of the precipitation signal. The inter-centennial patterns are largely weaker but negative versions of the inter-decadal patterns. For the net effect of all timescales, comparing the CMIP6 model output to the emulation output in Figs. 3 and 4 show good agreement with the strength and spatial distribution of the signals.

The global mean timeseries of the multi-model mean emulated fit is shown alongside the global mean timeseries of each ESM and the multi-model mean outputs in panels b (temperature) and d (precipitation) of Figure 5. The multi-model global mean response shown in panels a and b of Figure 5 matches the model mean calculated from the modelling output fairly well for both precipitation and temperature. As such, the multi-model mean of CMIP6 METEOR emulations is able to capture the global mean time evolution of the original ESM multi-model mean output well until 2100.

**Figure 3. METEOR GHG pattern from CMIP6.** GHG patterns obtained from the *abrupt4xCO2* scenario (CMIP6 model average) for temperature (left column) and precipitation (right column). Panels a and f show the CMIP6 model average in the 40 year average of the run between years 80 and 120. Panels b and g show the corresponding multi-model mean of the METEOR emulations for the scenario averaged over the same time period. Panels c and h, d and i and e and j show the contributions to the full emulations shown in b and g from the inter-annual, inter-decadal and inter-centennial modes, respectively.

**Figure 4. METEOR aerosol residuals pattern from CMIP6**. Residuals patterns obtained from the *histroical+SSP2-4.5* scenario (CMIP6 model average) for temperature (left column) and precipitation (right column). Panels a and g show the CMIP6 model average in the 40 year average of the run between years 1980 and 2020. Panels b and h show the corresponding multi-model mean of the METEOR emulations for the scenario averaged over the same time period. Panels c and i show the combined emulated GHG forcing pattern from all non-sulfate aerosol forcing. Panels d and j, e and k and f and l show the contributions to the full emulations shown in b and h from the inter-annual, inter-decadal and inter-centennial sulfate aerosol driven modes, respectively.

**Figure 5. METEOR GHG and aerosol residual reconstructions from CMIP6 models.** METEOR reconstruction for *abrupt4x-CO*<sub>2</sub> (a, c) and *historical+SSP2-4.5* (b,d) for global mean temperature (GMST, top row) and precipitation (GMP, bottom row) for each model (grey), the model mean (black) and the mean of the METEOR reconstructions (red for temperature and teal for precipitation). Panels b and d also include the mean reconstruction timeseries obtained using only the GHG pattern (stippled lines), the *abrupt4xCO*<sub>2</sub> reconstructions in panels a and c are identical with and without the aerosol pattern as only CO<sub>2</sub> is changing in this experiment.

# 3.2 Climate response reconstruction in out-of-sample scenarios



In the main study detailed here, only the future *SSP2-4.5* scenario is used in the training of METEOR, so other scenarios can be used as out-of-sample test cases. Figures 6, 7, 8, and 9 show the global mean and spatial pattern performance of the emulation applied to the ScenarioMIP scenarios. The timeseries plots (Figs. 6, 7) show both the response from greenhouse gas forcing alone (technically all forcers apart from sulfate aerosol), and the response from the model which includes the calibrated sulfate aerosol signal. The results demonstrate good performance of METEOR in out-of-sample scenarios, both in terms of the global mean and spatially resolved output for the multi-model mean of METEOR emulations compared to the CMIP6 multi-model mean. Supplementary Figs. B1 – B8 show similar global mean fits for each model included, and also show overall per model fidelity, though the time evolutions of some models are less well captured than others. In the global mean, the aerosol modes

**Figure 6. Global mean temperature reconstruction from METEOR vs CMIP6 data for various SSP scenarios.** Panels a–f show global mean *tas* (GMST) change for each model (grey), the model mean (black) and the mean of the METEOR reconstructions using GHG only (red dashed), and the combined GHG and aerosol patterns (red solid). The red plume shows the distribution of the full METEOR reconstructions. Panel f shows the year 2100 *tas* change with mean and modelling spread for the CMIP6 models (grey) and full METEOR reconstruction (red).

are able to capture the broad temporal dynamics of the global mean aerosol effect: cooling in the late 20th Century, and a reduced effect in the future in all scenarios except SSP3-7.0.

Panel f of Figures 6 and 7 shows year 2100 temperature and precipitation change mean and model spread for all SSPs included. For each scenario, METEOR is able to capture the future spread, though multi-model mean end of 21st century warming is slightly underestimated in the high mitigation *SSP1-2.6* experiment. Panels a-e in both figures include dashed lines showing the METEOR reconstruction results obtained using only the GHG patterns and forcing.



We can illustrate this visually by considering the multi-model mean spatial patterns of change in CMIP6 and the out-of-sample METEOR reconstructions, where the emulated amplitude and patterns of temperature and precipitation change are highly consistent for each scenario considered (Figs. 8, 9). The spatial bias is largest for SSP5-8.5 (the highest emissions scenario) for both temperature and precipitation. In the case of temperature there is cold bias for SSP5-8.5 and more of a warm bias for the low emissions and overshoot scenarios (SSP1-2.6 and SSP5-3.4-over). This may be an imprint of, or slight over-fitting to, the SSP2-4.5 scenario, which is colder than the former, and hotter than the two latter scenarios at the end of

Figure 7. Global mean precipitation reconstruction from METEOR vs CMIP6 data for various SSP scenarios. Panels a—f show global mean pr (GMP) change for each model (grey), the model mean (black) and the mean of the METEOR reconstructions using GHG only (teal dashed), and the combined GHG and aerosol patterns (teal solid). The teal plume shows the distribution of the full METEOR reconstructions. Panel f shows the year 2100 pr change with mean and modelling spread for the CMIP6 models (grey) and full METEOR reconstruction (teal).

the century. Note also that the colour scale in the bias plots (c, f and i) for temperature in Figure 8 has a smaller range, than the absolute value plots (a, b, d, e, g and h), whereas the scales are the same in Figure 9 mainly due to a stronger localised bias in the tropical pacific, especially for the *SSP5-85* scenario for precipitation. These patterns can also be seen in the spatial RMSE maps of Figure 14. In addition, supplementary Figs. B9, B10, B11 and B12 compare single model emulation for all the standard *SSP* scenarios for one of the models with the best METEOR emulations (NorESM2-MM) and one of the models with the worst METEOR emulations (CMCC-ESM2). Per model emulation to CMIP6 comparison maps for the *SSP5-3.4-over* scenarios for all models that we considered for that scenario are also shown in Figures B13 and B14.




For a more regional breakdown, Figs. 10 and 11 include CMIP6 regional averages for the last two decades of the 21st century for each of the *SSP* scenarios for 9 regions: Europe (0-45E, 37-75N), High Arctic (0-360E, 70-90N), Antarctic + Southern Ocean (0-360E, 55-90S), Tropics (0-360E, 20S-20N), South America (275-330E, 60S-15N), North America (200-310E, 10-70N), East Asia (65-150E, 5-60N), South East Asia and Australia (95-165E, 45S-5N), and Africa(0-60E, 40S-37N). METEOR's regional emulation of individual CMIP6 models is generally good for temperature across all regions (Fig. 10). Polar amplification is captured well, and, despite larger absolute differences, the relative performance of METEOR is best in

**Figure 8. METEOR temperature reconstruction pattern vs CMIP6 data for various SSP scenarios.** Panels in the first (a, d and g) and second (b, e and h) column show the difference between the mean of the first 50 years of the *historical* experiment (1850–1900) and the last two decades of the *SSP1-2.6*, *SSP5-8.5* and *SSP5-3.4-over* scenarios for the mean of the METEOR reconstructions and the CMIP6 model output, respectively. Panels in the third column (c, f and i) show the difference between the second and first column, illustrating the difference between the CMIP6 projections and the METEOR reconstruction.

the High Arctic region, while the fit is worse for the Southern Hemisphere high latitudes. Scenario-differences tend to remain constant regardless of their forcing strength, which increases METEOR's emulation skill for scenarios with higher forcings. For precipitation, the results are a bit more diverse, albeit consistent across all regions (Fig. 11). Again, the High Arctic region shows the best agreement between METEOR and CMIP6, while other regions show larger differences. For some regions and models, there is a contrasting direction of the precipitation response (i.e., opposite drying or wetting between METEOR and CMIP6), but this tends to be limited to low-magnitude precipitation changes.

**Figure 9. METEOR precipitation reconstruction pattern vs CMIP6 data for various SSP scenarios.** Panels in the first (a, d and g) and second (b, e and h) column show the the difference between the mean of the first 50 years of the *historical* experiment (1850–1900) and the last two decades of the *SSP1-2.6*, *SSP5-8.5* and *SSP5-3.4-over* scenarios for the mean of the METEOR reconstructions and the CMIP6 model output, respectively. Panels in the third column (c, f and i) show the difference between the second and first column, illustrating the difference between the CMIP6 projections and the METEOR reconstruction.

Figure 10. Regional CMIP6 vs. METEOR end of century temperature change. Regionally averaged temperature change (tas) since pre-industrial for CMIP6 output and METEOR per emulation for each model. Panels show average end of century temperature change for Europe (a), High Arctic (b), Antarctic + Southern Ocean (c), Tropics (d), South America (e), North America (f), East Asia (g), South East Asia and Australia (h) and Africa (i). Each circle shows the CMIP6 model result (y-axis) as function of the METEOR emulation (x-axis) for one model and scenario combination averaged over the region. Blue points show results for SSP1-2.6, green for SSP2-4.5, orange for SSP3-7.0, red for SSP5-8.5 and purple for SSP5-3.4-over. The black diagonal shows the line where model result and emulations are equal, and dashed lines show  $r^2$  distances from the linear regression per scenario.

Figure 11. Regional CMIP6 vs. METEOR end of century precipitation change. Regionally averaged precipitation change (pr) since pre-industrial for CMIP6 output and METEOR per emulation for each model. Panels show average end of century precipitation change for Europe (a), High Arctic (b), Antarctic + Southern Ocean (c), Tropics (d), South America (e), North America (f), East Asia (g), South East Asia and Australia (h) and Africa (i). Each circle shows the CMIP6 model result (y-axis) as function of the METEOR emulation (x-axis) for one model and scenario combination averaged over the region. Blue points show results for SSP1-2.6, green for SSP2-4.5, orange for SSP3-7.0, red for SSP5-8.5 and purple for SSP5-3.4-over. The black diagonal shows the line where model result and emulations are equal, and dashed lines show  $r^2$  distances from the linear regression per scenario.

### 3.3 Emulation of overshoot





A main motivation for the multiple timescale impulse response structure of METEOR is to allow for better emulation of overshoot including hysteresis behaviour in joint emulation of temperature and precipitation. We will therefore show some more detailed and per model results of the overshoot joint emulation through the example of the SSP5-3.4-over scenario.

Figure 12 shows the precipitation change as function of the temperature change globally and in each of 8 regions (Europe, High Arctic, Tropics, South America, North America, East Asia, South East Asia and Australia, and Africa). Globally, the emulation represents hysteresis effects evident in the CMIP6 ESMs well. At the regional scale, noise in the ESM simulations make this harder to assess, particularly in regions where the precipitation signal is relatively small. Nonetheless, in some regions such as the Tropics and South America, hysteresis features matching those of the ESMs are captured by METEOR.

Figure 13 show the end of century average maps of precipitation changes as function of temperature change in the SSP5-3.4-over scenario for each of the models. The results further illustrate METEOR's performance for the joint emulation on a per model basis. For some models like CESM2-WACCM and CanESM5, the regional co-evolution patterns seen in the emulation and the true model data are very similar, whereas for others there is a larger discrepancy. However, strong changes in the tropical Pacific are the dominant features in both models and emulation. The similarity of regional patterns of the temperature-dependent evolution of precipitation provides physics-based confidence in the joint evolution of temperature and precipitation in METEOR.

#### 3.4 Performance and error metrics

Here we show some selected error and performance metrics for the METEOR evaluation to scenarios.

Table 1 shows the Pearson's correlation coefficient and Root Mean Square Error (RMSE) for temperature and precipitation between the METEOR pattern reconstruction and CMIP6 end of century (2080–2100) change for each of the SSPs. In general, out-of-sample performance is better for temperature than for precipitation. Note that limited ESM simulations are available for the SSP5-3.4-over scenario. Performance in-sample (for SSP2-4.5) is better than out-of-sample performance, with the largest errors indicated in the high emission SSP5-8.5 scenario - but for all scenarios and variables considered, the correlation between spatial patterns of change exceeds 0.94. These fits are comparable to those reported for PRIME (see table 1 of Mathison et al. (2024)).

**Table 1.** Root Mean Square Error (RMSE, in K for *tas* and in kg m-2 s-1 for *pr*) and Pearson correlation coefficient (Pearson) between the METEOR pattern reconstruction and CMIP6 end of century (2080–2100) change.

| Metric  | Variable | SSP1-2.6 | SSP2-4.5 | SSP3-7.0 | SSP5-8.5 | SSP5-3.4-over |
|---------|----------|----------|----------|----------|----------|---------------|
| Pearson | tas      | 0.99     | 0.99     | 0.99     | 0.99     | 0.99          |
| Pearson | pr       | 0.97     | 0.99     | 0.99     | 0.96     | 0.94          |
| RMSE    | tas      | 0.13     | 0.15     | 0.34     | 0.50     | 0.32          |
| RMSE    | pr       | 5.7e-7   | 3.4e-7   | 7.1e-7   | 8.4e-7   | 13.1e-7       |

**Figure 12.** *SSP5-3.4-over* **regional precipitation versus temperature.** Global and regional results for precipitation change as a function of temperature change in the *SSP5-3.4-over* scenario. Each dot represents results for a specific model in a specific year averaged over the regions Global (a), Europe (a), High Arctic (b), Tropics (d), South America (e), North America (f), East Asia (g), Australia (h) and Africa (i) for the *SSP5-3.4-over* scenario. Lines show the trajectory of emulation results over the same regions for each of the models CESM2-WACCM (blue), CMCC-ESM2 (orange), CNRM-ESM2-1 (green), CanESM5 (red), MIROC-ES2L (grey) and MRI-ESM2-0 (purple).

**Figure 13.** *SSP5-3.4-over* **spatial precipitation versus temperature.** Spatial maps of end of century precipitation change as function of end of century temperature change for each model for the *SSP5-3.4-over* scenario. The left column (a, c, e, g, i and k) shows results of the METEOR emulation while the right column (b, d, f, h, j and l) shows the CMIP6 outputs directly.

Table 2. METEOR performance on NorESM-LM evaluated using and compared to the ClimateBench evaluation suite. (Watson-Parris et al., 2022; Watson-Parris, 2021). They compare normalised RMSE values obtained from comparing the average climate response over years 2080–2100 in model output and emulation (NRMSE<sub>s</sub>), normalising RMSE in the spatially averaged timeseries over the same years NRMSE<sub>g</sub> and taking the linear combination between the two, dubbed the total NRMSE (NRMSE<sub>t</sub> = NRMSE<sub>s</sub> +  $5 \cdot$  NRMSE<sub>g</sub>). \*The exact methods used to derive the ClimateBench provided output (Watson-Parris, 2021) for precipitation proved hard to replicate and hence to compare to our output, only the temperature outputs for SSP2-4.5 are compared directly to the ClimateBench output data itself. However, results were computed using the same metrics to the output for the ensemble member used for emulation taken from the CMIP6 database directly. Results for METEOR emulation errors for SSP1-2.6 and SSP3-7.0 were obtained in the same manner. The results for other emulations, internal variability and CMIP6 variability are taken directly from Watson-Parris et al. (2022). Note that the ClimateBench results are compared to the mean between three ensemble member outputs, hence some errors stemming from natural variability are dampened in the ClimateBench results and the result for SSP2-4.5 for METEOR compared to the other METEOR results.

|                     | NRMSE <sub>s</sub> tas | $NRMSE_g$ tas | $NRMSE_t$ tas | $NRMSE_s$ pr | $\mathrm{NRMSE}_g$ pr | $\mathrm{NRMSE}_t$ pr |
|---------------------|------------------------|---------------|---------------|--------------|-----------------------|-----------------------|
| METEOR ssp126       | 0.165                  | 0.161         | 0.969         | 1.978        | 0.100                 | 2.480                 |
| METEOR ssp245       | 0.053                  | 0.042         | 0.262         | 2.041*       | 0.111*                | 2.594*                |
| METEOR ssp370       | 0.118                  | 0.082         | 0.527         | 2.139        | 0.133                 | 2.804                 |
| Gaussian Process CB | 0.109                  | 0.074         | 0.478         | 2.341        | 0.341                 | 4.048                 |
| Neural Network CB   | 0.107                  | 0.044         | 0.327         | 2.128        | 0.209                 | 3.175                 |
| Random Forest CB    | 0.108                  | 0.058         | 0.400         | 2.524        | 0.502                 | 5.035                 |
| Pattern Scaling CB  | 0.080                  | 0.048         | 0.320         | 2.006        | 0.331                 | 3.662                 |
| Variability CB      | 0.052                  | 0.072         | 0.414         | 1.350        | 0.268                 | 2.691                 |
| CMIP6 CB            | 0.258                  | 0.177         | 1.141         | 1.994        | 0.389                 | 3.940                 |

Table 2 shows comparisons to the ClimateBench evaluation suite (Watson-Parris et al., 2022; Watson-Parris, 2021), which combines temporal and spatial NRMSE values for emulations to the *SSP2-4.5* run of NorESM-LM specifically for several different emulation techniques. Results show that METEOR emulations are on par with the other emulation techniques. Note that NorESM-LM is not part of the CMIP6 multi modelled ensemble considered in the rest of this article, as it did not have available outputs for the *SSP5-8.5* scenario. Similar metrics for the models used in our CMIP6 model ensemble are listed in table supplementary tables B2–B5. As overshoot is a special focus for METEOR, Table 3 shows the ClimateBench results for each of the models we have considered that ran the *SSP5-3.4-over* scenario.




Finally, Figure 14 shows the maps of spatial RMSE for the end of the century emulation of CMIP6 models for four scenarios: *SSP1-2.6*, *SSP2-4.5*, *SSP3-7.0* and *SSP5-8.5*. The *SSP2-4.5* scenario overall shows the smallest RMSE, which is expected as it is METEOR's training scenario. For the higher emission scenarios *SSP3-7.0* and *SSP5-8.5*, the high Arctic and the Southern Ocean shows larger RMSE for temperature, while for precipitation, largest RMSE are found over the tropical oceans. In the *SSP1-2.6* scenario, the RMSE for temperature shows overall the same pattern both for temperature and precipitation, but the former is much attenuated over the high Arctic. This indicates that, by training to *SSP2-4.5*, some non-linear Arctic ocean

**Table 3.** METEOR performance for all models evaluated using the Climate bench evaluation metrics. Note that comparison is to single ensemble members, so variability driven errors are included. \*Precipitation metrics for MIROC-E2SL had issues.

|                         | $NRMSE_s$ tas | $NRMSE_g$ tas | $NRMSE_t$ tas | $\mathrm{NRMSE}_{s}$ pr | $NRMSE_g$ pr | $\mathrm{NRMSE}_t$ pr |
|-------------------------|---------------|---------------|---------------|-------------------------|--------------|-----------------------|
| CESM2-WACCM ssp534-over | 0.229         | 0.207         | 1.264         | 0.074                   | 0.009        | 0.117                 |
| CMCC-ESM2 ssp534-over   | 0.144         | 0.120         | 0.743         | 0.058                   | 0.009        | 0.104                 |
| CNRM-ESM2-1 ssp534-over | 0.289         | 0.251         | 1.542         | 0.070                   | 0.013        | 0.137                 |
| CanESM5 ssp534-over     | 0.205         | 0.180         | 1.107         | 0.098                   | 0.010        | 0.150                 |
| MIROC-E2SL ssp534-over  | 0.271         | 0.212         | 1.331         | *                       | *            | *                     |
| MRI-ESM2-0 ssp534-over  | 0.170         | 0.075         | 0.543         | 0.069                   | 0.010        | 0.119                 |

**Figure 14. Spatial RMSE.** End of century spatial RMSE for temperature (top row) and precipitation (bottom row) in each of the standard scenarios *SSP1-2.6* (a, e), *SSP2-4.5* (b, f) *SSP3-7.0* (c, g) and *SSP5-8.5* (d, h).

or sea-ice dynamics are not captured by METEOR in the higher emission out-of-sample scenarios for temperature, while the differences tend to scale more linearly for precipitation.

### 4 Conclusions, discussion and outlook


Here, we have presented the METEOR (Multivariate Emulation of Time-Evolving and Overlapping Responses) v1.0 emulator framework for spatially resolved climate impacts. The framework allows for the reproduction of time-evolving response to a range of radiative forcers, allowing for the simulation of hysteresis dynamics and forcer-dependent responses.

We showed results of its training and application to CMIP6 models for experiments *abrupt4x-CO*<sub>2</sub>, *piControl*, *histori*cal, SSP2-4.5, SSP1-2.6, SSP3-7.0, SSP5-8.5 and SSP5-3.4-over for annual mean temperature (tas) and precipitation (pr). METEOR displays good overall performance for both in- and out-of-sample applications and though performance is (unsurprisingly) slightly worse for out-of-sample applications (particularly for precipitation), the amplitude and spatial pattern of multi-model response is well captured for all scenarios considered. Beyond the demonstrations here, the model can easily be trained on new modelling output and to emulate new and unknown scenarios and reasonable accuracy can be expected.

The model is structured as an importable Python library with example Jupyter notebooks that demonstrate basic usage and reproduce the figures in this article. This should make the model accessible, useful and extendable for new uses for the wider climate research community.

### 4.1 Discussion







Though METEOR shows good fits, including the ability to fit hysteresis, it has limitations. For instance, the model assumes that increasing patterns and timescales are associated with increasingly longer and non-overlapping timescales. In practice, there might be several pattern effects occurring on more closely related timescales, or on timescales which are shorter or longer than those considered here. Allowing for additional modes in testing did not lead to notable increases in performance. This is partly a function of data limits in training data. In particular, the number of models which have the shortest possible timescale of 1 year indicate that a shorter mode might yield better results. However, timescales of less than 1 year cannot be represented given the annual mean data used for training in this study, and training simulations are generally run for 150–250 years, hence shorter or longer timescales cannot be meaningfully fitted with these data. As with any emulator, performance is limited by the availability of training data; capturing multi-millennial timescale responses requires training data to unambiguously simulate those timescales. Fitting shorter (sub-annual) timescales requires an extension of the methodology considered here and any meaningful approach would require a treatment of the seasonal cycle and internal variability, which is planned for future releases of METEOR. Further, gaining high confidence in shorter timescale responses would require large initial condition ensembles, which would allow the assessment of forced response given presence of internal variability with large impacts on the estimation of faster response timescales (Rugenstein et al., 2016). In general, the number of independent forcer responses and timescales could be improved with ESM datasets which isolate the effect of independent forcers.

In METEORv1.0, we assume that the anomaly between the full target model response and the synthetic simulated green-house gas only response is due to sulfate aerosol forcing. However, in practice, anomalies will also be caused by any errors in the reconstruction of the synthetic GHG response, by other forcers which are not explicitly represented or by potential nonlinear interactions between forcers. The model framework is sufficiently flexible to test these possibilities, but data from current CMIP archives is too limited to conduct these refinements for the full range of models considered here. There exist experiments which could be used to provide additional information for some models, such as the DAMIP experiments (Gillett et al., 2016): ssp245-aer ssp245-GHG, ssp245-CO2, ssp245-stratO3, which we wish to exploit in the future. However, for this demonstration, we prioritise showing METEOR's performance using only experiments which have been performed by most CMIP6 models, and therefore demonstrating METEOR's versatility. Hence, we isolate the effect of GHG and sulfate aerosol forcing leaning on results from the literature (Myhre et al., 2017; Samset et al., 2018, 2019; Zhao et al., 2019; Monerie et al., 2022; Persad et al., 2023; Wilcox et al., 2023) including the assessments of the last IPCC cycle (Forster et al., 2021), which

highlight the strength, peculiarity and uncertainty of the aerosol and specifically sulfate aerosol forcing as a first order modifier from the pure GHG-driven response.

The strength and usefulness of an emulator such as METEOR lies particularly in its ability to produce impact results rapidly. In this paper we have focused on annual mean values of temperature, but for impact studies, seasonal cycle and extreme value data might be more useful, and some emulators already have extensions that allow for monthly or seasonal output (Nath et al., 2022; Schöngart et al., 2024; Tebaldi et al., 2022; Nath et al., 2024). Future versions of METEOR will seek to represent these additional dimensions, noting that METEOR's variable agnostic setup can facilitate, for example, training on seasonally varying data, i.e. building separate patterns for January temperatures, February temperatures, summer temperatures or maximum yearly temperatures.

As capturing hysteresis for modelling overshoot scenarios is an explicit motivation for the METEOR methodology, the apparent success of its fit to the *SSP5-3.4-over* scenario is reassuring. However, as only very few models have full available training data for this scenario, the robustness of METEOR for overshoot scenarios should be explored further. We note in particular that the experiment used to identify GHG timescales and patterns (*abrupt4x-CO*<sub>2</sub>) only simulates the effects of a permanent increase in concentrations, so that in particular timescales associated with negative emissions from processes such as Carbon Dioxide Removal might differ. Experiments in CMIP7, such as flat10MIP (Sanderson et al., 2024) will provide assessments of potential asymmetries between responses to positive and negative emissions in Earth System Models, which could be used as additional training data for METEOR.

For the modelling presented in this paper, we chose a combination of *historical* and *SSP2-4.5* to fit the residual sulfate aerosol patterns. In Figure 8, though quite small, we note a slight hot bias for *SSP1-2.6* and a slight cold bias for *SSP5-8.5*. This makes sense as the *SSP2-4.5* temperature trajectory lies between the two. This implies that there may be, on average, some nonlinear temperature response in CMIP models which is not captured in the pulse-response logic exploited here, and will likely also mean that fits are probably slightly more accurate when applied to scenarios that are not too far off from the scenario used for the residual fitting. In practice the biases here are small, and fits overall are good in the range considered, but if the intended application of METEOR is to scenarios in a very particular temperature trajectory range, choosing a residual to fit to that is relatively centrally located for that range is advisable. For the purposes of this paper as well as most ordinary applications to CMIP6 data, we consider that *SSP2-4.5* is a reasonable choice. Similarly, the aerosol profile in the future may have low emissions of sulfate aerosol, while nitrate aerosol emissions might increase (Adams et al., 2001; Liao and Seinfeld, 2005; Bauer et al., 2007; Liao et al., 2009; Bellouin et al., 2011; Hauglustaine et al., 2014). The forcing responses to nitrate aerosols are currently not well constrained, and can be highly dependent on the location and height of the emissions source (Aamaas et al., 2016). In addition to possible biases from all non-CO2 forcers that are treated as GHG, but which may in practice feed into the residual pattern, nitrate aerosol effects might be a particular bias to look out for in future projections.

### 4.2 Outlook






The potential applications of METEOR are not limited to mean temperature and precipitation. It can be trained and used to model climate change response in any variable for which a strong connection between climate change forcing and variable

evolution can be assumed. Examples can be impact related variables such as extreme value impact indicators (Quilcaille et al., 2022, 2023; Sillmann et al., 2013) or other climatological variables such as humidity, radiative balance or soil moisture. Further development would extend the model to more directly resolve impacts such as crop yields, human heat stress and sea level rise. However, we expect such indicators will require additional modules.








The scheme used to obtain the GHG response from *abrupt4x-CO*<sub>2</sub> can be employed in the same way for any forcer for which there is data for an abrupt step-change response. A collection of such experiments are contained in PDRMIP (Myhre et al., 2017, 2022). However, PDRMIP was performed with what are now somewhat outdated modelling versions. METEOR can be applied to that dataset, or a similar updated one, to obtain separate per forcer timescales and patterns, allowing for further decomposition of spatial forcer response as a function of a wider range of species.

The scheme for anomaly calculations that we employed to obtain the aerosol signal may also be performed for multiple forcers if data from experiments where only a single forcer is changed are available. For instance, the DAMIP (The Detection and Attribution Model Intercomparison Project) (Gillett et al., 2016) and RAMIP (Regional Aerosol Model Intercomparison Project) (Wilcox et al., 2023) datasets offer such data, the latter of which can even separate the forcing response from aerosols depending on their region of origin, which may in fact produce different results both in terms of spatial pattern and strength of climate results (Wilcox et al., 2023).

The timescales and patterns that METEOR obtains for any particular model, though mostly operational and for emulation purposes, may also indicate aspects of the underlying physics of the model, and comparison between parameters obtained for emulation of different models may itself provide indicators for understanding ESM differences (such as the role of fast and slow feedback processes in observed climate and implications for future climate commitments).

The emissions-to-forcing pipeline from the C-SCM is currently an integral part of METEOR. However, coupling METEOR to a similar pipeline from a different simple climate model, or allowing it to run directly from input forcing data is a logical future development goal, allowing uncertainty in the emissions-forcing pipeline to be decoupled from the forcing-pattern component. An alternative would be a higher level of integration within a simple climate model - such that the METEOR forcing-pattern mapping becomes an integral part of a wider model, including, for example, ecosystem components which could evolve as a function of regional climate as simulated within METEOR.

Currently, METEOR does not have any representation of natural variability, nor is there native support for producing probabilistic output which spans uncertainty in either ESM training model, or in fitting uncertainty for a single model. Each of these would be useful for impact applications, and would be logical extensions for future development. Probabilistic spatial information is implemented using various techniques for the MESMER framework (Beusch et al., 2020; Schöngart et al., 2024; Quilcaille et al., 2022, 2023). Goals for future versions of METEOR include probabilistic implementations in which a set of plausible METEOR configurations can be produced for a given ESM emulation, such that an ensemble of simulations can then provide risk guidance for climate impacts. Testing the robustness of timescales and patterns across ensemble runs of the same model would be beneficial in achieving this goal. Using emulations based on existing ensemble spread as in Schwaab et al. (2024) or including some variability into the emulator process pipeline itself similar to what has been done on a global scale for the FAIR simple climate model (Bouabid et al., 2024) are both possible approaches to this.

METEORv1.0 is tailored towards annual data. However, for many applications, monthly or higher time resolution outputs are desirable. The methodology can be fairly easily generalised to monthly output (for example, by expanding the dimensionality of the spatial dimension to include different months), but further testing is required to ensure the realism of emulated timeseries. Additional modules are planned to model dominant modes of natural variability, and potentially extreme value indicators.

There may be indications that extremely fast (sub-yearly, near instantaneous), and very slow timescales (millennial) are not captured by the current setup. Aiming to fit the former could be a targeted extension, but the latter might require output from longer runs than are currently available for most models. The assumptions made of exponentially longer timescale lengths might also not be ideal, as local optima for several shorter or comparable timescales may not be found. A more thorough investigation of this may be needed, but Appendix A does include a discussion and investigation of the fits as a function of number of timescales included.

We provide METEOR as an open tool which we hope can be of wider benefit to the community, contributing to a wider body of emulation tools each of which provides unique advantages with ample scope for intercomparisons, coupling and ensemble studies. Finally, we look forward to community development of the METEOR platform to provide better integration into ecosystems of fast climate modelling tools which can be increasingly used in applications which require rapid turnaround, such as simulating regional climate impacts in societal models.

Code availability. Code is openly available on github at https://github.com/benmsanderson/METEOR under the Apache-2.0 license at the v1.0.2 tag with doi https://doi.org/10.5281/zenodo.15732955 Sanderson et al. (2025)

Data availability. Emissions input data are from the Reduced Complexity Model Intercomparison Nicholls et al. (2020), data at Nicholls and

Gieseke (2019). CMIP6 data are available through the Earth System Grid Federation (ESGF; Cinquini et al., 2014) or via the zarrstore googleapi (https://storage.googleapis.com/cmip6/cmip6-zarr-consolidated-stores.csv). Additional per model figures are available at https://doi.org/10.5281/zenoc
Steinert and Sanderson (2025).

### Appendix A: Fitness depending on number of timescales



The METEOR model can be trained with an arbitrary number of timescales  $\tau_k$ . In this paper we have shown results for a setup which assumes three timescales for each forcer. However, the user can specify to train and use an arbitrary number of timescales and corresponding spatial patterns. Figure A1 shows the effect of increasing the number of timescales on the emulation fits. Panels a and d show each model's results as connected points, the RMSE of the fit to the *abrupt4x-CO*<sub>2</sub> as the number of timescales increase. Panels b an e show these values only for the multi model mean. The fitness increases substantially for both temperature and precipitation when going from one to two timescales, and a further slight increase is observed when increasing to three timescales.

Panels c and f of Fig. A1, show the RMSE model mean fit for temperature and precipitation with varying numbers of both GHG and residual aerosol timescales to the SSP2-4.5 experiment. Annual values per model for global mean reconstruction versus model output for each combination of GHG and aerosol timescales are shown in Figures A2 (temperature) and A3 (precipitation). From this, we observe that a combination of three timescales for both aerosols and GHG provides a reasonable balance. When a single aerosol timescale is used, we observe that the fit actually deteriorates as GHG timescales are increased from two to three. We believe that this is a consequence of the range constraints for timescales. With one or two very slow GHG timescales, the residual pattern includes timescale signals which do not quite fit the SSP2-4.5 scenario. With only one very short timescale with which to compensate for this, the residual can not compensate. This does not mean that sulfate aerosols have very long timescale mechanisms, but rather should serve to caution the user on the interpretability of the emulator outputs. Since the sulfate aerosol timescales and patterns are based on residual signals, they also include and have in them information which is just to do with the lack of accuracy of the abrupt4x-CO<sub>2</sub> based GHG patterns and timescales to accurately map the effect of everything else that happens in the model in the historical and SSP2-4.5 scenario runs. This compensating factor lead us to find it reasonable to use three timescales also for the aerosol patterns, and this shows the best overall fits by these measures.

Figure A4 shows the contributions of the different timescales and how they combine. Panels a (temperature) and c (precipitation) show the global mean time evolution of the reconstruction associated with each timescale, and we can see that all GHG patterns are associate with positive contributions, whereas the aerosol residual patterns on inter-annual and inter-decadal scales have negative contributions, with the inter-centennial pattern giving a positive contribution. Panels b (temperature) and d (precipitation) show how the total reconstruction changes as subsequent patterns are added starting with the shortest timescale for GHG, adding longer GHG timescales first, before adding the aerosol patterns in order of timescale length.

### Appendix B: Results for single model emulation







Here we include results per individual model emulated. Table B1 lists all models included and the timescales obtained for the main emulations of them. The model selection criteria was mainly one of convenience, including models that had data available from zarrr store CMIP6 google-api (https://storage.googleapis.com/cmip6/cmip6-zarr-consolidated-stores.csv) for all of the experiments *piControl*, *abrupt4x-CO*<sub>2</sub>, *SSP1-2.6*, *SSP2-4.5*, *SSP3-7.0* and , *SSP5-8.5*. From this set, some models were excluded for various issues with the data available. Tables B2–B5 show ClimateBench-style (Watson-Parris et al., 2022) error scores for all model and experiment combinations.

Global mean reconstructions per model for each of the four SSP experiments are shown in Figures B1–B3 (temperature) and Figures B5–B7 (precipitation). In addition, a smaller selection of models which also had data for *SSP5-3.4-over* were used to evaluate the performance of the emulation in overshoot scenarios. Global mean reconstruction results for these are shown individually in Figures B4 (temperature) and B8 (precipitation).

We also show plots that illustrate the single model spatial performance of METEOR, by comparing end of 21st century emulation and CMIP6 data for all the standard scenarios for one model for which METEOR had some of the overall best

**Figure A1. Evaluation of METEOR reconstruction as function of number of timescales**. Panels a and d show the evolution of the RMSE parameter for temperature and precipitation respectively in the reconstruction of the *abrupt4x-CO*<sub>2</sub> experiment as a function of increased number of timescales for each individual CMIP6 model. Panels b and e show mean of the same numbers. Panels c and f show the same mean RMSE but applied to the reconstruction of the *SSP2-4.5* experiment and including a varying number of aerosol timescales modelled from residual.

**Figure A2. Evaluation of** *tas* **for varying number of timescales**. Columns left to right show increasing numbers of aerosol timescales from 0 to 4, while rows from top to bottom show increasing numbers of GHG timescales from 1 to 4. Each point represents a global mean value for temperature for one model in the original CMIP6 data versus the METEOR reconstruction for *SSP2-4.5*. Model mean overall RMSE and mean absolute error (MAE) are displayed for each timescale combination.

**Figure A3. Evaluation of** *pr* **for varying number of timescales**. Columns left to right show increasing numbers of aerosol timescales from 0 to 4, while rows from top to bottom show increasing numbers of GHG timescales from 1 to 4. Each point represents a global mean value for temperature for one model in the original CMIP6 data versus the METEOR reconstruction for *SSP2-4.5*. Model mean overall RMSE and mean absolute error (MAE) are displayed for each timescale combination.

**Figure A4. Multi-model mean per timescale contributions.** With a model trained with three GHG and three aerosol timescales, we show the model mean global reconstruction contributions from each timescale pattern when emulating *SSP2-4.5*. Panels a (temperature) and c (precipitation) show the separate contributions from the inter-annual GHG (red), inter-decadal GHG (orange), inter-centennial GHG (green), inter-annual aerosol (dark blue), inter-decadal aerosol (medium blue) and inter-centennial aerosol (turquoise) overlayed on top of the original CMIP6 multi model data (grey). Panels b (temperature) and d (precipitation) show the result of summing the patterns in this order.

performance (NorESM2-MM) in Figs. B9 and B11 and for one for which METEOR had some of the overall worst performance (CMCC-ESM2) in Figs. B10 and B12. For the *SSP5-3.4-over* scenario, we display the end of century emulation map comparison for all models in Figs. B13 and B14, showing more of a spread in performance.

Author contributions. BMS designed the initial concept for the emulator and provided help throughout its development. SB contributed to the the initial concept and design. MS was the primary developer of the software infrastructure. NJS produced and made layouts for the figures and the code flow schematic. NJS, BMS, SB and MS all contributed to writing and reviewing all parts of the article.

Competing interests. The authors declare that they have no conflict of interest.


**Figure B1.** GMST original data (grey), fit reconstruction for only GHG (orange) and full reconstruction (purple) for single models, one model per row. The results were obtained using *SSP2-4.5* to fit the aerosol sulfate residual. From left to right the columns show *SSP1-2.6* (out-of-sample), *SSP2-4.5* (in-sample), *SSP3-7.0* (out-of-sample) and *SSP5-8.5* (out-of-sample) fits.

**Figure B2.** GMST original data (grey), fit reconstruction for only GHG (orange) and full reconstruction (purple) for single models, one model per row. The results were obtained using *SSP2-4.5* to fit the aerosol sulfate residual. From left to right the columns show *SSP1-2.6* (out-of-sample), *SSP2-4.5* (in-sample), *SSP3-7.0* (out-of-sample) and *SSP5-8.5* (out-of-sample) fits.

**Table B1.** METEOR pattern scaling timescales  $\tau$  split into inter-annual (0–10 years), inter-decadal (10–100 years) and inter-centennial (100-1000 years) timescales provided for each CMIP6 model, for temperature and precipitation and greenhouse gas (GHG) and residuals (aer) responses, respectively. Note that in METEORv1.0.1, the respective timescales are forced to fit within the specified ranges.

| Earth System Model |         |         | Temperature |         |         |         |         |         | Precipitation |         |         |                    |
|--------------------|---------|---------|-------------|---------|---------|---------|---------|---------|---------------|---------|---------|--------------------|
|                    |         | GHG     |             |         | aer     |         |         | GHG     |               |         | aer     |                    |
|                    | $	au_1$ | $	au_2$ | $	au_3$     | $	au_1$ | $	au_2$ | $	au_3$ | $	au_1$ | $	au_2$ | $	au_3$       | $	au_1$ | $	au_2$ | $\overline{	au_3}$ |
| ACCESS-CM2         | 1.0     | 12.7    | 100.0       | 1.1     | 73.4    | 998.1   | 6.4     | 100.0   | 1000.0        | 1.2     | 95.4    | 719.1              |
| ACCESS-ESM1-5      | 1.2     | 12.9    | 100.0       | 1.8     | 100.0   | 1000.0  | 1.0     | 10.0    | 996.0         | 2.4     | 100.0   | 1000.0             |
| AWI-CM-1-1-MR      | 1.0     | 10.0    | 490.0       | 2.3     | 18.9    | 102.8   | 3.0     | 10.0    | 1000.0        | 2.0     | 10.0    | 163.4              |
| BCC-CSM2-MR        | 1.0     | 10.0    | 974.1       | 1.0     | 10.0    | 100.0   | 1.0     | 10.0    | 1000.0        | 1.0     | 11.3    | 100.0              |
| CAMS-CSM1-0        | 1.0     | 10.9    | 980.2       | 1.0     | 10.0    | 100.0   | 2.0     | 12.0    | 1000.0        | 1.0     | 100.0   | 1000.0             |
| CAS-ESM2-0         | 1.0     | 12.2    | 298.0       | 1.4     | 100.0   | 1000.0  | 4.7     | 31.6    | 837.3         | 1.0     | 100.0   | 1000.0             |
| CESM2              | 1.1     | 10.0    | 993.4       | 2.3     | 100.0   | 1000.0  | 1.0     | 10.0    | 995.4         | 3.9     | 100.0   | 1000.0             |
| CMCC-CM2-SR5       | 1.0     | 10.0    | 123.3       | 1.0     | 100.0   | 538.3   | 6.9     | 100.0   | 1000.0        | 8.6     | 59.5    | 1000.0             |
| CMCC-ESM2          | 1.0     | 10.0    | 212.9       | 1.0     | 59.7    | 221.6   | 7.2     | 100.0   | 1000.0        | 10.0    | 41.0    | 179.6              |
| CNRM-CM6-1         | 1.0     | 15.0    | 100.0       | 1.0     | 100.0   | 1000.0  | 7.7     | 100.0   | 1000.0        | 1.0     | 100.0   | 1000.0             |
| CNRM-CM6-1-HR      | 1.1     | 15.1    | 1000.0      | 1.0     | 100.0   | 1000.0  | 1.0     | 10.5    | 687.8         | 1.0     | 100.0   | 1000.0             |
| CNRM-ESM2-1        | 1.2     | 10.0    | 999.9       | 1.0     | 100.0   | 1000.0  | 10.0    | 99.0    | 711.3         | 1.0     | 100.0   | 1000.0             |
| CanESM5            | 1.2     | 13.0    | 429.9       | 5.7     | 100.0   | 1000.0  | 10.0    | 12.8    | 1000.0        | 4.5     | 99.1    | 281.6              |
| EC-Earth3-Veg      | 1.0     | 10.9    | 100.0       | 1.0     | 100.0   | 1000.0  | 4.9     | 100.0   | 1000.0        | 1.0     | 100.0   | 1000.0             |
| FGOALS-f3-L        | 1.0     | 10.0    | 975.6       | 1.6     | 10.0    | 100.0   | 6.7     | 100.0   | 116.8         | 7.2     | 16.9    | 100.0              |
| GFDL-ESM4          | 1.0     | 10.0    | 999.9       | 1.6     | 100.0   | 1000.0  | 10.0    | 100.0   | 791.5         | 1.0     | 100.0   | 1000.0             |
| GISS-E2-1-H        | 1.0     | 10.0    | 939.4       | 2.5     | 100.0   | 1000.0  | 10.0    | 24.0    | 100.0         | 1.3     | 100.0   | 1000.0             |
| INM-CM4-8          | 1.0     | 14.2    | 964.7       | 1.0     | 10.0    | 1000.0  | 8.1     | 100.0   | 1000.0        | 9.3     | 100.0   | 1000.0             |
| INM-CM5-0          | 1.0     | 10.7    | 1000.0      | 1.0     | 100.0   | 1000.0  | 10.0    | 49.5    | 100.0         | 10.0    | 100.0   | 1000.0             |
| KACE-1-0-G         | 1.0     | 10.0    | 939.4       | 2.5     | 100.0   | 1000.0  | 10.0    | 24.9    | 100.0         | 1.3     | 100.0   | 1000.0             |
| MCM-UA-1-0         | 1.0     | 10.0    | 759.0       | 1.0     | 45.0    | 1000.0  | 2.9     | 18.9    | 999.9         | 4.3     | 100.0   | 1000.0             |
| MIROC-ES2L         | 1.0     | 14.2    | 964.7       | 1.0     | 10.0    | 1000.0  | 8.1     | 100.0   | 1000.0        | 9.3     | 100.0   | 1000.0             |
| MPI-ESM1-2-HR      | 1.0     | 10.7    | 1000.0      | 1.0     | 100.0   | 1000.0  | 10.0    | 49.5    | 100.0         | 10.0    | 100.0   | 1000.0             |
| NorESM2-MM         | 1.0     | 10.0    | 1000.0      | 1.2     | 100.0   | 1000.0  | 4.2     | 13.3    | 1000.0        | 1.0     | 15.1    | 100.0              |
| UKESM1-0-LL        | 1.6     | 100.0   | 1000.0      | 2.5     | 99.9    | 999.1   | 10.0    | 16.1    | 100.0         | 5.2     | 100.0   | 1000.0             |

Acknowledgements. The European Union through the Horizon 2020 project PROVIDE (grant agreement No. 101003687) and the Horizon Europe Research and Innovation programme project DIAMOND (grant no. 101081179) funded this work. We thank Bjørn Hallvard Samset and Camilla Weum Stjern for clarifying discussions on the topic of aerosols. We thank anonymous reviewer 1 and reviewer 2, Yann Quilcaille, for their constructive criticisms and valuable suggestions for improving the manuscript in the review process. ChatGPT was used in the initial formulation of the introduction all of which has since been checked and heavily rewritten.

**Table B2.** METEOR performance for all models (A–CE) evaluated using the Climate bench evaluation metrics. Note that comparison is to single ensemble members, so variability driven errors are included.

|                      | $NRMSE_s$ tas | $NRMSE_g$ tas | $NRMSE_t$ tas | $NRMSE_s$ pr | $NRMSE_g$ pr | $\mathrm{NRMSE}_t$ pr |
|----------------------|---------------|---------------|---------------|--------------|--------------|-----------------------|
| ACCESS-CM2 ssp126    | 0.114         | 0.037         | 0.298         | 0.079        | 0.003        | 0.096                 |
| ACCESS-CM2 ssp245    | 0.027         | 0.017         | 0.113         | 0.024        | 0.003        | 0.041                 |
| ACCESS-CM2 ssp370    | 0.087         | 0.068         | 0.429         | 0.097        | 0.004        | 0.118                 |
| ACCESS-CM2 ssp585    | 0.080         | 0.018         | 0.171         | 0.122        | 0.007        | 0.155                 |
| ACCESS-ESM1-5 ssp126 | 0.176         | 0.090         | 0.624         | 0.076        | 0.005        | 0.101                 |
| ACCESS-ESM1-5 ssp245 | 0.033         | 0.025         | 0.159         | 0.029        | 0.003        | 0.045                 |
| ACCESS-ESM1-5 ssp370 | 0.080         | 0.043         | 0.294         | 0.113        | 0.003        | 0.130                 |
| ACCESS-ESM1-5 ssp585 | 0.069         | 0.034         | 0.237         | 0.101        | 0.005        | 0.128                 |
| AWI-CM-1-1-MR ssp126 | 0.121         | 0.054         | 0.392         | 0.057        | 0.002        | 0.069                 |
| AWI-CM-1-1-MR ssp245 | 0.032         | 0.042         | 0.240         | 0.038        | 0.003        | 0.051                 |
| AWI-CM-1-1-MR ssp370 | 0.058         | 0.036         | 0.239         | 0.067        | 0.005        | 0.090                 |
| AWI-CM-1-1-MR ssp585 | 0.092         | 0.047         | 0.325         | 0.105        | 0.011        | 0.161                 |
| BCC-CSM2-MR ssp126   | 0.133         | 0.072         | 0.492         | 0.067        | 0.003        | 0.085                 |
| BCC-CSM2-MR ssp245   | 0.050         | 0.037         | 0.236         | 0.040        | 0.003        | 0.053                 |
| BCC-CSM2-MR ssp370   | 0.159         | 0.114         | 0.727         | 0.065        | 0.006        | 0.094                 |
| BCC-CSM2-MR ssp585   | 0.087         | 0.037         | 0.272         | 0.079        | 0.005        | 0.106                 |
| CAMS-CSM1-0 ssp126   | 0.152         | 0.126         | 0.785         | 0.060        | 0.007        | 0.094                 |
| CAMS-CSM1-0 ssp245   | 0.078         | 0.062         | 0.388         | 0.024        | 0.008        | 0.065                 |
| CAMS-CSM1-0 ssp370   | 0.080         | 0.062         | 0.390         | 0.064        | 0.007        | 0.101                 |
| CAMS-CSM1-0 ssp585   | 0.153         | 0.135         | 0.830         | 0.098        | 0.016        | 0.177                 |
| CAS-ESM2-0 ssp126    | 0.112         | 0.051         | 0.366         | 0.058        | 0.005        | 0.083                 |
| CAS-ESM2-0 ssp245    | 0.025         | 0.031         | 0.180         | 0.020        | 0.004        | 0.040                 |
| CAS-ESM2-0 ssp370    | 0.081         | 0.050         | 0.329         | 0.060        | 0.015        | 0.136                 |
| CAS-ESM2-0 ssp585    | 0.065         | 0.042         | 0.273         | 0.080        | 0.010        | 0.129                 |
| CESM2 ssp126         | 0.146         | 0.055         | 0.421         | 0.077        | 0.005        | 0.100                 |
| CESM2 ssp245         | 0.033         | 0.040         | 0.232         | 0.026        | 0.005        | 0.051                 |
| CESM2 ssp370         | 0.120         | 0.047         | 0.357         | 0.076        | 0.014        | 0.145                 |
| CESM2 ssp585         | 0.087         | 0.019         | 0.182         | 0.079        | 0.005        | 0.102                 |

**Table B3.** METEOR performance for all models (CM–Can) evaluated using the Climate bench evaluation metrics. Note that comparison is to single ensemble members, so variability driven errors are included.

|                      | $NRMSE_s$ tas | $NRMSE_g$ tas | $NRMSE_t$ tas | $\mathrm{NRMSE}_{s}$ pr | $\mathrm{NRMSE}_g$ pr | $\mathrm{NRMSE}_t$ pr |
|----------------------|---------------|---------------|---------------|-------------------------|-----------------------|-----------------------|
| CMCC-CM2-SR5 ssp126  | 0.133         | 0.063         | 0.449         | 0.066                   | 0.004                 | 0.088                 |
| CMCC-CM2-SR5 ssp245  | 0.039         | 0.038         | 0.229         | 0.022                   | 0.005                 | 0.046                 |
| CMCC-CM2-SR5 ssp370  | 0.271         | 0.151         | 1.023         | 0.096                   | 0.021                 | 0.199                 |
| CMCC-CM2-SR5 ssp585  | 0.170         | 0.081         | 0.574         | 0.078                   | 0.011                 | 0.132                 |
| CMCC-ESM2 ssp126     | 0.159         | 0.083         | 0.574         | 0.060                   | 0.007                 | 0.097                 |
| CMCC-ESM2 ssp245     | 0.045         | 0.033         | 0.211         | 0.024                   | 0.005                 | 0.049                 |
| CMCC-ESM2 ssp370     | 0.264         | 0.149         | 1.007         | 0.065                   | 0.020                 | 0.166                 |
| CMCC-ESM2 ssp585     | 0.177         | 0.085         | 0.601         | 0.083                   | 0.013                 | 0.146                 |
| CNRM-CM6-1 ssp126    | 0.155         | 0.069         | 0.499         | 0.056                   | 0.004                 | 0.078                 |
| CNRM-CM6-1 ssp245    | 0.052         | 0.052         | 0.311         | 0.026                   | 0.005                 | 0.051                 |
| CNRM-CM6-1 ssp370    | 0.112         | 0.074         | 0.483         | 0.062                   | 0.007                 | 0.095                 |
| CNRM-CM6-1 ssp585    | 0.073         | 0.048         | 0.312         | 0.061                   | 0.005                 | 0.089                 |
| CNRM-CM6-1-HR ssp126 | 0.116         | 0.050         | 0.367         | 0.053                   | 0.003                 | 0.068                 |
| CNRM-CM6-1-HR ssp245 | 0.048         | 0.021         | 0.153         | 0.029                   | 0.003                 | 0.045                 |
| CNRM-CM6-1-HR ssp370 | 0.111         | 0.039         | 0.306         | 0.051                   | 0.006                 | 0.080                 |
| CNRM-CM6-1-HR ssp585 | 0.127         | 0.053         | 0.392         | 0.060                   | 0.007                 | 0.094                 |
| CNRM-ESM2-1 ssp126   | 0.139         | 0.064         | 0.460         | 0.061                   | 0.004                 | 0.082                 |
| CNRM-ESM2-1 ssp245   | 0.086         | 0.060         | 0.384         | 0.025                   | 0.005                 | 0.049                 |
| CNRM-ESM2-1 ssp370   | 0.093         | 0.055         | 0.368         | 0.072                   | 0.004                 | 0.093                 |
| CNRM-ESM2-1 ssp585   | 0.064         | 0.029         | 0.208         | 0.063                   | 0.005                 | 0.087                 |
| CanESM5 ssp126       | 0.152         | 0.064         | 0.471         | 0.087                   | 0.006                 | 0.117                 |
| CanESM5 ssp245       | 0.027         | 0.029         | 0.170         | 0.026                   | 0.003                 | 0.043                 |
| CanESM5 ssp370       | 0.102         | 0.076         | 0.480         | 0.099                   | 0.003                 | 0.115                 |
| CanESM5 ssp585       | 0.104         | 0.039         | 0.297         | 0.120                   | 0.004                 | 0.142                 |

**Table B4.** METEOR performance for all models (E–I) evaluated using the Climate bench evaluation metrics. Note that comparison is to single ensemble members, so variability driven errors are included.

|                      | $NRMSE_s$ tas | $NRMSE_g$ tas | $NRMSE_t$ tas | $\mathrm{NRMSE}_s$ pr | $\mathrm{NRMSE}_g$ pr | $\mathrm{NRMSE}_t$ pr |
|----------------------|---------------|---------------|---------------|-----------------------|-----------------------|-----------------------|
| EC-Earth3-Veg ssp126 | 0.997         | 0.992         | 5.958         | 0.066                 | 0.009                 | 0.109                 |
| EC-Earth3-Veg ssp245 | 0.994         | 0.989         | 5.938         | 0.098                 | 0.006                 | 0.131                 |
| EC-Earth3-Veg ssp370 | 0.987         | 0.986         | 5.917         | 0.080                 | 0.006                 | 0.111                 |
| EC-Earth3-Veg ssp585 | 0.986         | 0.982         | 5.894         | 0.085                 | 0.009                 | 0.131                 |
| FGOALS-f3-L ssp126   | 0.164         | 0.067         | 0.499         | 0.084                 | 0.009                 | 0.131                 |
| FGOALS-f3-L ssp245   | 0.034         | 0.061         | 0.339         | 0.022                 | 0.009                 | 0.068                 |
| FGOALS-f3-L ssp370   | 0.079         | 0.048         | 0.319         | 0.087                 | 0.008                 | 0.128                 |
| FGOALS-f3-L ssp585   | 0.111         | 0.064         | 0.433         | 0.085                 | 0.014                 | 0.153                 |
| GFDL-ESM4 ssp126     | 0.118         | 0.071         | 0.471         | 0.070                 | 0.006                 | 0.098                 |
| GFDL-ESM4 ssp245     | 0.044         | 0.052         | 0.303         | 0.036                 | 0.006                 | 0.064                 |
| GFDL-ESM4 ssp370     | 0.086         | 0.049         | 0.331         | 0.065                 | 0.008                 | 0.105                 |
| GFDL-ESM4 ssp585     | 0.057         | 0.032         | 0.218         | 0.074                 | 0.008                 | 0.113                 |
| GISS-E2-1-H ssp126   | 0.180         | 0.058         | 0.472         | 0.081                 | 0.011                 | 0.138                 |
| GISS-E2-1-H ssp245   | 0.067         | 0.048         | 0.308         | 0.045                 | 0.003                 | 0.061                 |
| GISS-E2-1-H ssp370   | 0.107         | 0.050         | 0.357         | 0.111                 | 0.021                 | 0.215                 |
| GISS-E2-1-H ssp585   | 0.102         | 0.066         | 0.430         | 0.115                 | 0.009                 | 0.158                 |
| INM-CM4-8 ssp126     | 0.208         | 0.099         | 0.701         | 0.083                 | 0.005                 | 0.106                 |
| INM-CM4-8 ssp245     | 0.040         | 0.053         | 0.303         | 0.024                 | 0.004                 | 0.045                 |
| INM-CM4-8 ssp370     | 0.095         | 0.050         | 0.344         | 0.073                 | 0.010                 | 0.123                 |
| INM-CM4-8 ssp585     | 0.114         | 0.058         | 0.407         | 0.076                 | 0.009                 | 0.121                 |
| INM-CM5-0 ssp126     | 0.995         | 0.994         | 5.966         | 1.199                 | 0.967                 | 6.035                 |
| INM-CM5-0 ssp245     | 0.993         | 0.992         | 5.952         | 1.184                 | 0.957                 | 5.971                 |
| INM-CM5-0 ssp370     | 0.990         | 0.989         | 5.935         | 1.172                 | 0.946                 | 5.904                 |
| INM-CM5-0 ssp585     | 0.988         | 0.987         | 5.924         | 1.164                 | 0.940                 | 5.863                 |

**Table B5.** METEOR performance for all models (K–Z) evaluated using the Climate bench evaluation metrics. Note that comparison is to single ensemble members, so variability driven errors are included.

|                      | $NRMSE_s$ tas | $NRMSE_g$ tas | $NRMSE_t$ tas | $NRMSE_s$ pr | $NRMSE_g$ pr | $\mathrm{NRMSE}_t$ pr |
|----------------------|---------------|---------------|---------------|--------------|--------------|-----------------------|
| KACE-1-0-G ssp126    | 0.283         | 0.069         | 0.628         | 0.067        | 0.003        | 0.084                 |
| KACE-1-0-G ssp245    | 0.061         | 0.045         | 0.285         | 0.029        | 0.004        | 0.051                 |
| KACE-1-0-G ssp370    | 0.114         | 0.056         | 0.396         | 0.074        | 0.010        | 0.126                 |
| KACE-1-0-G ssp585    | 0.114         | 0.056         | 0.393         | 0.078        | 0.007        | 0.114                 |
| MCM-UA-1-0 ssp126    | 0.115         | 0.072         | 0.475         | 0.064        | 0.004        | 0.083                 |
| MCM-UA-1-0 ssp245    | 0.059         | 0.042         | 0.270         | 0.028        | 0.004        | 0.050                 |
| MCM-UA-1-0 ssp370    | 0.114         | 0.051         | 0.369         | 0.067        | 0.006        | 0.096                 |
| MCM-UA-1-0 ssp585    | 0.118         | 0.068         | 0.457         | 0.083        | 0.009        | 0.126                 |
| MIROC-ES2L ssp126    | 0.158         | 0.070         | 0.505         | 0.057        | 0.005        | 0.084                 |
| MIROC-ES2L ssp245    | 0.062         | 0.016         | 0.142         | 0.022        | 0.002        | 0.034                 |
| MIROC-ES2L ssp370    | 0.101         | 0.069         | 0.444         | 0.056        | 0.004        | 0.075                 |
| MIROC-ES2L ssp585    | 0.096         | 0.061         | 0.403         | 0.068        | 0.010        | 0.120                 |
| MPI-ESM1-2-HR ssp126 | 0.181         | 0.090         | 0.633         | 0.064        | 0.003        | 0.079                 |
| MPI-ESM1-2-HR ssp245 | 0.041         | 0.030         | 0.189         | 0.022        | 0.003        | 0.037                 |
| MPI-ESM1-2-HR ssp370 | 0.093         | 0.050         | 0.344         | 0.053        | 0.004        | 0.074                 |
| MPI-ESM1-2-HR ssp585 | 0.075         | 0.023         | 0.188         | 0.060        | 0.005        | 0.085                 |
| NorESM2-MM ssp126    | 0.996         | 0.995         | 5.970         | 1.252        | 0.985        | 6.177                 |
| NorESM2-MM ssp245    | 0.993         | 0.992         | 5.953         | 1.240        | 0.976        | 6.121                 |
| NorESM2-MM ssp370    | 0.990         | 0.989         | 5.934         | 1.229        | 0.969        | 6.075                 |
| NorESM2-MM ssp585    | 0.987         | 0.986         | 5.916         | 1.210        | 0.955        | 5.984                 |
| UKESM1-0-LL ssp126   | 0.136         | 0.051         | 0.394         | 0.070        | 0.004        | 0.091                 |
| UKESM1-0-LL ssp245   | 0.061         | 0.051         | 0.314         | 0.046        | 0.003        | 0.063                 |
| UKESM1-0-LL ssp370   | 0.076         | 0.028         | 0.217         | 0.102        | 0.005        | 0.125                 |
| UKESM1-0-LL ssp585   | 0.075         | 0.012         | 0.137         | 0.102        | 0.006        | 0.133                 |

**Figure B3.** GMST original data (grey), fit reconstruction for only GHG (orange) and full reconstruction (purple) for single models, one model per row. The results were obtained using *SSP2-4.5* to fit the aerosol sulfate residual. From left to right the columns show *SSP1-2.6* (out-of-sample), *SSP2-4.5* (in-sample), *SSP3-7.0* (out-of-sample) and *SSP5-8.5* (out-of-sample) fits.

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

**Figure B4.** GMST original data (grey), fit reconstruction for only GHG (orange) and full reconstruction (purple) for single models that had reasonable data quality for the overshoot scenario *SSP5-3.4-over*. The results were obtained using *SSP2-4.5* to fit the aerosol sulfate residual. Each sub-plot shows results for a single model.

- Bellouin, N., Rae, J., Jones, A., Johnson, C., Haywood, J., and Boucher, O.: Aerosol forcing in the Climate Model Intercomparison Project (CMIP5) simulations by HadGEM2-ES and the role of ammonium nitrate, Journal of Geophysical Research: Atmospheres, 116, https://doi.org/https://doi.org/10.1029/2011JD016074, 2011.
- Beusch, L., Gudmundsson, L., and Seneviratne, S. I.: Emulating Earth system model temperatures with MESMER: from global mean temperature trajectories to grid-point-level realizations on land, Earth System Dynamics, 11, 139–159, https://doi.org/10.5194/esd-11-139-2020, 2020.

- Beusch, L., Nicholls, Z., Gudmundsson, L., Hauser, M., Meinshausen, M., and Seneviratne, S. I.: From emission scenarios to spatially resolved projections with a chain of computationally efficient emulators: coupling of MAGICC (v7.5.1) and MESMER (v0.8.3), Geoscientific Model Development, 15, 2085–2103, https://doi.org/10.5194/gmd-15-2085-2022, 2022.
- Bouabid, S., Sejdinovic, D., and Watson-Parris, D.: FaIRGP: A Bayesian Energy Balance Model for Surface Temperatures Emulation, Journal of Advances in Modeling Earth Systems, 16, e2023MS003926, https://doi.org/https://doi.org/10.1029/2023MS003926, e2023MS003926 2023MS003926, 2024.
- Cinquini, L., Crichton, D., Mattmann, C., Harney, J., Shipman, G., Wang, F., Ananthakrishnan, R., Miller, N., Denvil, S., Morgan, M., Pobre,
   Z., Bell, G. M., Doutriaux, C., Drach, R., Williams, D., Kershaw, P., Pascoe, S., Gonzalez, E., Fiore, S., and Schweitzer, R.: The Earth System Grid Federation: An open infrastructure for access to distributed geospatial data, Future Gener. Comput. Syst., 36, 400–417, https://esgf-node.llnl.gov, 2014.
- Collins, M., Knutti, R., Arblaster, J., Dufresne, J.-L., Fichefet, T., Friedlingstein, P., Gao, X., Gutowski, W., Johns, T., Krinner, G., Shongwe, M., Tebaldi, C., Weaver, A., and Wehner, M.: Long-term Climate Change: Projections, Commitments and Irreversibility, in: Climate
   Change 2013: The Physical Science Basis. Contribution of Working Group I to the Fifth Assessment Report of the Intergovernmental Panel on Climate Change, edited by Stocker, T., Qin, D., Plattner, G.-K., Tignor, M., Allen, S., Boschung, J., Nauels, A., Xia, Y., Bex,

**Figure B5.** GMP original data (grey), fit reconstruction for only GHG (orange) and full reconstruction (purple) for single models, one model per row. The results were obtained using *SSP2-4.5* to fit the aerosol sulfate residual. From left to right the columns show *SSP1-2.6* (out-of-sample), *SSP2-4.5* (in-sample), *SSP3-7.0* (out-of-sample) and *SSP5-8.5* (out-of-sample) fits.

**Figure B6.** GMP original data (grey), fit reconstruction for only GHG (orange) and full reconstruction (purple) for single models, one model per row. The results were obtained using *SSP2-4.5* to fit the aerosol sulfate residual. From left to right the columns show *SSP1-2.6* (out-of-sample), *SSP2-4.5* (in-sample), *SSP3-7.0* (out-of-sample) and *SSP5-8.5* (out-of-sample) fits.

**Figure B7.** GMP original data (grey), fit reconstruction for only GHG (orange) and full reconstruction (purple) for single models, one model per row. The results were obtained using *SSP2-4.5* to fit the aerosol sulfate residual. From left to right the columns show *SSP1-2.6* (out-of-sample), *SSP2-4.5* (in-sample), *SSP3-7.0* (out-of-sample) and *SSP5-8.5* (out-of-sample) fits.

V., and Midgley, P., book section 12, p. 1029–1136, Cambridge University Press, Cambridge, United Kingdom and New York, NY, USA, ISBN ISBN 978-1-107-66182-0, https://doi.org/10.1017/CBO9781107415324.024, 2013.

Dietz, S., Rising, J., Stoerk, T., and Wagner, G.: Economic impacts of tipping points in the climate system, Proceedings of the National Academy of Sciences, 118, e2103081118, https://doi.org/10.1073/pnas.2103081118, 2021.


Dunne, J. P., Hewitt, H. T., Arblaster, J., Bonou, F., Boucher, O., Cavazos, T., Durack, P. J., Hassler, B., Juckes, M., Miyakawa, T., Mizielinski, M., Naik, V., Nicholls, Z., O'Rourke, E., Pincus, R., Sanderson, B. M., Simpson, I. R., and Taylor, K. E.: An evolving Coupled Model Intercomparison Project phase 7 (CMIP7) and Fast Track in support of future climate assessment, EGUsphere, 2024, 1–51, https://doi.org/10.5194/egusphere-2024-3874, 2024.

Eyring, V., Bony, S., Meehl, G. A., Senior, C. A., Stevens, B., Stouffer, R. J., and Taylor, K. E.: Overview of the Coupled Model Intercomparison Project Phase 6 (CMIP6) experimental design and organization, Geoscientific Model Development, 9, 1937–1958, https://doi.org/10.5194/gmd-9-1937-2016, 2016.

Ferrari, L., Carlino, A., Gazzotti, P., Tavoni, M., and Castelletti, A.: From optimal to robust climate strategies: expanding integrated assessment model ensembles to manage economic, social, and environmental objectives, Environmental Research Letters, 17, 084 029, 2022.

**Figure B8.** GMP original data (grey), fit reconstruction for only GHG (orange) and full reconstruction (purple) for single models that had reasonable data quality for the overshoot scenario *SSP5-3.4-over*. The results were obtained using *SSP2-4.5* to fit the aerosol sulfate residual. Each sub-plot shows results for a single model.

Forster, P., Storelvmo, T., Armour, K., Collins, W., Dufresne, J.-L., Frame, D., Lunt, D., Mauritsen, T., Palmer, M., Watanabe, M., Wild, M., and Zhang, H.: The Earth's Energy Budget, Climate Feedbacks, and Climate Sensitivity, in: Climate Change 2021: The Physical Science Basis. Contribution of Working Group I to the Sixth Assessment Report of the Intergovernmental Panel on Climate Change, edited by Masson-Delmotte, V., Zhai, P., Pirani, A., Connors, S. L., Péan, C., Berger, S., Caud, N., Chen, Y., Goldfarb, L., Gomis, M. I., Huang, M., Leitzell, K., Lonnoy, E., Matthews, J. B. R., Maycock, T. K., Waterfield, T., Yelekçi, O., Yu, R., and Zhou, B., book section 7, pp. 923–1054, Cambridge University Press, Cambridge, UK and New York, NY, USA, https://doi.org/10.1017/9781009157896.009, 2021.

Forster, P. M., Smith, C., Walsh, T., Lamb, W. F., Lamboll, R., Cassou, C., Hauser, M., Hausfather, Z., Lee, J.-Y., Palmer, M. D., von Schuckmann, K., Slangen, A. B. A., Szopa, S., Trewin, B., Yun, J., Gillett, N. P., Jenkins, S., Matthews, H. D., Raghavan, K., Ribes, A., Rogelj, J., Rosen, D., Zhang, X., Allen, M., Aleluia Reis, L., Andrew, R. M., Betts, R. A., Borger, A., Broersma, J. A., Burgess, S. N., Cheng, L., Friedlingstein, P., Domingues, C. M., Gambarini, M., Gasser, T., Gütschow, J., Ishii, M., Kadow, C., Kennedy, J., Killick, R. E., Krummel, P. B., Liné, A., Monselesan, D. P., Morice, C., Mühle, J., Naik, V., Peters, G. P., Pirani, A., Pongratz, J., Minx, J. C., Rigby, M., Rohde, R., Savita, A., Seneviratne, S. I., Thorne, P., Wells, C., Western, L. M., van der Werf, G. R., Wijffels, S. E., Masson-Delmotte, V., and Zhai, P.: Indicators of Global Climate Change 2024: annual update of key indicators of the state of the climate system and human influence, Earth System Science Data, 17, 2641–2680, https://doi.org/10.5194/essd-17-2641-2025, 2025.



Gillett, N. P., Shiogama, H., Funke, B., Hegerl, G., Knutti, R., Matthes, K., Santer, B. D., Stone, D., and Tebaldi, C.: The Detection and Attribution Model Intercomparison Project (DAMIP v1.0) contribution to CMIP6, Geoscientific Model Development, 9, 3685–3697, https://doi.org/10.5194/gmd-9-3685-2016, 2016.

Good, P., Lowe, J. A., Andrews, T., Wiltshire, A., Chadwick, R., Ridley, J. K., Menary, M. B., Bouttes, N., Dufresne, J. L., Gregory, J. M., Schaller, N., and Shiogama, H.: Nonlinear regional warming with increasing CO2 concentrations, Nature Climate Change, 5, https://doi.org/10.1038/nclimate2498, 2015.

**Figure B9. NorESM2-MM temperature emulation comparison.** NorESM2-MM was found to be one of the models for which METEOR has the best overall fit. Here we display the emulated patterns (a, d, g, j) compared to the CMIP6 data (b, e, h, k) and the difference between them (c, f, i, l) for the temperature emulation of the standard *SSP* scenarios at the end of the century; *SSP1-2.6* (a, b, c), *SSP2-4.5* (d, e, f), *SSP3-7.0* (g, h, i) and *SSP5-8.5* (j, k, l) for this model.

Hauglustaine, D. A., Balkanski, Y., and Schulz, M.: A global model simulation of present and future nitrate aerosols and their direct radiative forcing of climate, Atmospheric Chemistry and Physics, 14, 11 031–11 063, https://doi.org/10.5194/acp-14-11031-2014, 2014.

Herger, N., Sanderson, B. M., and Knutti, R.: Improved pattern scaling approaches for the use in climate impact studies, Geophysical Research Letters, 42, 3486–3494, https://doi.org/10.1002/2015GL063569, 2015.

Hulme, M., Wigley, T. M. L., Barrow, E. andRaper, S., Centella, A., Smith, S., and Chipanshi, A.: Using a Climate Scenario Generator for Vulnerability and Adaptation Assessments, in: MAGICC and SCENGEN Version 2.4 Workbook and CD-ROM, edited by Lim, B. and Smith, J., Reprint, East Anglia: University of East Anglia. Climatic Research Unit, https://opensky.ucar.edu/islandora/object/books% 3A487, 2000.


Huntingford, C. et al.: Climate sensitivity analysis of forest growth uncertainties, Global Ecology and Biogeography, 9, 359–374, 2000.

**Figure B10. CMCC-ESM2 temperature emulation comparison.** CMCC-ESM2 was found to be one of the models for which METEOR has the worst overall fit. Here we display the emulation patterns (a, d, g, j) compared to the CMIP6 data (b, e, h, k) and the difference between them (c, f, i, l) for the temperature emulation of the standard *SSP* scenarios at the end of the century; *SSP1-2.6* (a, b, c), *SSP2-4.5* (d, e, f), *SSP3-7.0* (g, h, i) and *SSP5-8.5* (j, k, l) for this model.

IPCC: Climate Change 2021: The Physical Science Basis, Cambridge University Press, Cambridge, United Kingdom and New York, NY, USA, 2021.

IPCC: Climate Change 2022: Impacts, Adaptation and Vulnerability. Contribution of Working Group II to the Sixth Assessment Report of the Intergovernmental Panel on Climate Change, Cambridge University Press, Cambridge, UK and New York, NY, USA, https://doi.org/10.1017/9781009325844, 2022.

James Rising, Marco Tedesco, F. P. D. A. S.: The missing risks of climate change, Nature, 610, 643–651, https://doi.org/10.1038/s41586-022-05243-6, 2022.


Jonko, A. K., Shell, K. M., Sanderson, B. M., and Danabasoglu, G.: Climate feedbacks in CCSM3 under changing CO 2 forcing. Part II: Variation of climate feedbacks and sensitivity with forcing, Journal of Climate, 26, 2784–2795, 2013.

**Figure B11. NorESM2-MM precipitation emulation comparison.** NorESM2-MM was found to be one of the models for which METEOR has the best overall fit. Here we display the emulation patterns (a, d, g, j) compared to the CMIP6 data (b, e, h, k) and the difference between them (c, f, i, l) for the precipitation emulation of the standard *SSP* scenarios at the end of the century; *SSP1-2.6* (a, b, c), *SSP2-4.5* (d, e, f), *SSP3-7.0* (g, h, i) and *SSP5-8.5* (j, k, l) for this model.

**Figure B12.** CMCC-ESM2 precipitation emulation comparison. CMCC-ESM2 was found to be one of the models for which METEOR has the worst overall fit. Here we display the emulation patterns (a, d, g, j) compared to the CMIP6 data (b, e, h, k) and the difference between them (c, f, i, l) for the precipitation emulation of the standard *SSP* scenarios at the end of the century; *SSP1-2.6* (a, b, c), *SSP2-4.5* (d, e, f), *SSP3-7.0* (g, h, i) and *SSP5-8.5* (j, k, l) for this model.

Figure B13. SSP5-3.4-over temperature emulation comparison. SSP5-3.4-over spatial temperature emulation patterns (a, d, g, j, m, p) compared to the CMIP6 data (b, e, h, k, n, q) and the difference between them (c, f, i, l, o, r, u) at the end of the century for each of the models that had sufficient SSP5-3.4-over data available.

**Figure B14.** *SSP5-3.4-over* **precipitation emulation comparison.** *SSP5-3.4-over* spatial precipitation emulation patterns (a, d, g, j, m, p) compared to the CMIP6 data (b, e, h, k, n, q) and the difference between them (c, f, i, l, o, r, u) at the end of the century for each of the models that had sufficient *SSP5-3.4-over* data available.

- Kikstra, J. S., Waidelich, P., James Rising, D. Y., Hope, C., and Brierley, C. M.: The social cost of carbon dioxide under climate-economy feedbacks and temperature variability, Environmental Research Letters, 16, https://doi.org/DOI 10.1088/1748-9326/ac1d0b, 2021.
- 620 Liao, H. and Seinfeld, J. H.: Global impacts of gas-phase chemistry-aerosol interactions on direct radiative forcing by anthropogenic aerosols and ozone, Journal of Geophysical Research: Atmospheres, 110, https://doi.org/https://doi.org/10.1029/2005JD005907, 2005.
  - Liao, H., Zhang, Y., Chen, W.-T., Raes, F., and Seinfeld, J. H.: Effect of chemistry-aerosol-climate coupling on predictions of future climate and future levels of tropospheric ozone and aerosols, Journal of Geophysical Research: Atmospheres, 114, https://doi.org/https://doi.org/10.1029/2008JD010984, 2009.
- Mansfield, L. A., Nowack, P. J., Kasoar, M., Everitt, R. G., Collins, W. J., and Voulgarakis, A.: Predicting global patterns of long-term climate change from short-term simulations using machine learning, npj Climate and Atmospheric Science, 3, https://doi.org/10.1038/s41612-020-00148-5, 2020.
  - Mathison, C. T., Burke, E., Kovacs, E., Munday, G., Huntingford, C., Jones, C., Smith, C., Steinert, N., Wiltshire, A., Gohar, L., and Varney, R.: A rapid application emissions-to-impacts tool for scenario assessment: Probabilistic Regional Impacts from Model patterns and Emissions (PRIME), EGUsphere [preprint], pp. 1–28, https://doi.org/https://doi.org/10.5194/egusphere-2023-2932, 2024.




- Mitchell, T. D.: Pattern Scaling: An Examination of the Accuracy of the Technique for Describing Future Climates, Climatic Change, 60, 217–242, https://doi.org/10.1023/A:1026035305597, 2003.
- Monerie, P.-A., Wilcox, L. J., and Turner, A. G.: Effects of Anthropogenic Aerosol and Greenhouse Gas Emissions on Northern Hemisphere Monsoon Precipitation: Mechanisms and Uncertainty, Journal of Climate, 35, 2305 2326, https://doi.org/10.1175/JCLI-D-21-0412.1, 2022.
- Myhre, G., Forster, P. M., Samset, B. H., Hodnebrog, Ø., Sillmann, J., Aalbergsjø, S. G., Andrews, T., Boucher, O., Faluvegi, G., Fläschner, D., Iversen, T., Kasoar, M., Kharin, V., Kirkevåg, A., Lamarque, J.-F., Olivié, D., Richardson, T. B., Shindell, D., Shine, K. P., Stjern, C. W., Takemura, T., Voulgarakis, A., and Zwiers, F.: PDRMIP: A Precipitation Driver and Response Model Intercomparison Project—Protocol and Preliminary Results, Bulletin of the American Meteorological Society, 98, 1185 1198, https://doi.org/10.1175/BAMS-D-16-0019.1, 2017.
- Myhre, G., Samset, B., Forster, P. M., Hodnebrog, Ø., Sandstad, M., Mohr, C. W., Sillmann, J., Stjern, C. W., Andrews, T., Boucher, O., Faluvegi, G., Iversen, T., Lamarque, J.-F., Kasoar, M., Kirkevåg, A., Kramer, R., Liu, L., Mülmenstädt, J., Olivié, D., Quaas, J., Richardson, T. B., Shawki, D., Shindell, D., Smith, C., Stier, P., Tang, T., Takemura, T., Voulgarakis, A., and Watson-Parris, D.: Scientific data from precipitation driver response model intercomparison project, Scientific Data, 9, 2052–4463, https://doi.org/10.1038/s41597-022-01194-9, 2022.
- Nath, S., Lejeune, Q., Beusch, L., Seneviratne, S. I., and Schleussner, C.-F.: MESMER-M: an Earth system model emulator for spatially resolved monthly temperature, Earth System Dynamics, 13, 851–877, https://doi.org/10.5194/esd-13-851-2022, 2022.
- Nath, S., Carreau, J., Kornhuber, K., Pfleiderer, P., Schleussner, C.-F., and Naveau, P.: MERCURY: A fast and versatile multi-resolution based global emulator of compound climate hazards, https://arxiv.org/abs/2501.04018, 2024.
- 650 Nicholls, Z. and Gieseke, R.: RCMIP Phase 1 Data (v2.0.0), https://doi.org/10.5281/zenodo.4016613, https://doi.org/https://doi.org/10.5281/zenodo.4016613, dataset, 2019.
  - Nicholls, Z., Meinshausen, M., Lewis, J., Corradi, M. R., Dorheim, K., Gasser, T., Gieseke, R., Hope, A. P., Leach, N. J., McBride, L. A., Quilcaille, Y., Rogelj, J., Salawitch, R. J., Samset, B. H., Sandstad, M., Shiklomanov, A., Skeie, R. B., Smith, C. J., Smith, S. J., Su, X., Tsutsui, J., Vega-Westhoff, B., and Woodard, D. L.: Reduced Complexity Model Intercomparison

Project Phase 2: Synthesizing Earth System Knowledge for Probabilistic Climate Projections, Earth's Future, 9, e2020EF001900, https://doi.org/https://doi.org/10.1029/2020EF001900, e2020EF001900 2020EF001900, 2021.

Model Development, 13, 5175-5190, https://doi.org/10.5194/gmd-13-5175-2020, 2020.


- Nicholls, Z. R. J., Meinshausen, M., Lewis, J., Gieseke, R., Dommenget, D., Dorheim, K., Fan, C.-S., Fuglestvedt, J. S., Gasser, T., Golüke, U., Goodwin, P., Hartin, C., Hope, A. P., Kriegler, E., Leach, N. J., Marchegiani, D., McBride, L. A., Quilcaille, Y., Rogelj, J., Salawitch, R. J., Samset, B. H., Sandstad, M., Shiklomanov, A. N., Skeie, R. B., Smith, C. J., Smith, S., Tanaka, K., Tsutsui, J., and Xie, Z.: Reduced Complexity Model Intercomparison Project Phase 1: introduction and evaluation of global-mean temperature response, Geoscientific
- O'Neill, B. C., Tebaldi, C., van Vuuren, D. P., Eyring, V., Friedlingstein, P., Hurtt, G., Knutti, R., Kriegler, E., Lamarque, J.-F., Lowe, J., Meehl, G. A., Moss, R., Riahi, K., and Sanderson, B. M.: The Scenario Model Intercomparison Project (ScenarioMIP) for CMIP6, Geoscientific Model Development, 9, 3461–3482. https://doi.org/10.5194/gmd-9-3461-2016, 2016.
- Persad, G., Samset, B. H., Wilcox, L. J., Allen, R. J., Bollasina, M. A., Booth, B. B. B., Bonfils, C., Crocker, T., Joshi, M., Lund, M. T., Marvel, K., Merikanto, J., Nordling, K., Undorf, S., van Vuuren, D. P., Westervelt, D. M., and Zhao, A.: Rapidly evolving aerosol emissions are a dangerous omission from near-term climate risk assessments, Environmental Research: Climate, 2, 032 001, https://doi.org/10.1088/2752-5295/acd6af, 2023.
- Proistosescu, C. and Huybers, P. J.: Slow climate mode reconciles historical and model-based estimates of climate sensitivity, Science Advances, 3, e1602 821, 2017.
  - Quilcaille, Y., Gudmundsson, L., Beusch, L., Hauser, M., and Seneviratne, S. I.: Showcasing MESMER-X: Spatially Resolved Emulation of Annual Maximum Temperatures of Earth System Models, Geophysical Research Letters, 49, e2022GL099012, https://doi.org/10.1029/2022GL099012, e2022GL099012 2022GL099012, 2022.
- Quilcaille, Y., Gudmundsson, L., and Seneviratne, S. I.: Extending MESMER-X: a spatially resolved Earth system model emulator for fire weather and soil moisture, Earth System Dynamics, 14, 1333–1362, 2023.
  - Rugenstein, M. A., Gregory, J. M., Schaller, N., Sedláček, J., and Knutti, R.: Multiannual ocean–atmosphere adjustments to radiative forcing, Journal of Climate, 29, 5643–5659, 2016.
- Samset, B. H., Sand, M., Smith, C. J., Bauer, S. E., Forster, P. M., Fuglestvedt, J. S., Osprey, S., and Schleussner, C.-F.: Climate Impacts From a Removal of Anthropogenic Aerosol Emissions, Geophysical Research Letters, 45, 1020–1029, https://doi.org/https://doi.org/10.1002/2017GL076079, 2018.
  - Samset, B. H., Lund, M. T., Bollasina, M., Myhre, G., and Wilcox, L.: Emerging Asian aerosol patterns, Nature Geoscience, 12, 582–584, https://doi.org/10.1038/s41561-019-0424-5, 2019.
  - Sanderson, B., Sandstad, M., and normansteinert: benmsanderson/METEOR: v1.0.1 (v1.0.1), Zenodo, https://doi.org/10.5281/zenodo.14967116, 2025.
- Sanderson, B. M., Brovkin, V., Fisher, R., Hohn, D., Ilyina, T., Jones, C., Koenigk, T., Koven, C., Li, H., Lawrence, D., et al.: flat10MIP: An emissions-driven experiment to diagnose the climate response to positive, zero, and negative CO2 emissions, EGUsphere, 2024, 1–39, 2024.
  - Sandstad, M., Aamaas, B., Johansen, A. N., Lund, M. T., Peters, G. P., Samset, B. H., Sanderson, B. M., and Skeie, R. B.: CICERO Simple Climate Model (CICERO-SCM v1.1.1) an improved simple climate model with a parameter calibration tool, Geosci. Model Dev., 17, 6589–6625, 2024.

- Santer, B. D., Wigley, T. M., Schlesinger, M. E., and Mitchell, J. F.: Developing Climate Scenarios from Equilibrium GCM Results, Max-Planck-Institut für Meteorologie, https://api.semanticscholar.org/CorpusID:127017512, 1990.
- Schöngart, S., Gudmundsson, L., Hauser, M., Pfleiderer, P., Lejeune, Q., Nath, S., Seneviratne, S. I., and Schleussner, C.-F.: Introducing the MESMER-M-TPv0.1.0 module: spatially explicit Earth system model emulation for monthly precipitation and temperature, Geoscientific Model Development, 17, 8283–8320, https://doi.org/10.5194/gmd-17-8283-2024, 2024.
  - Schwaab, J., Hauser, M., Lamboll, R. D., Beusch, L., Gudmundsson, L., Quilcaille, Y., Lejeune, Q., Schöngart, S., Schleussner, C.-F., Nath, S., Rogelj, J., Nicholls, Z., and Seneviratne, S. I.: Spatially resolved emulated annual temperature projections for overshoot pathways, Scientific Data. 11, https://doi.org/10.1038/s41597-024-04122-1, 2024.
- Shiogama, H., Emori, S., Takahashi, K., Nagashima, T., Ogura, T., Nozawa, T., and Takemura, T.: Emission Scenario Dependency of Precipitation on Global Warming in the MIROC3.2 Model, Journal of Climate, 23, 2404 2417, https://doi.org/10.1175/2009JCLI3428.1, 2010.
  - Sillmann, J., Kharin, V. V., Zwiers, F. W., Zhang, X., and Bronaugh, D.: Climate extremes indices in the CMIP5 multimodel ensemble: Part 2. Future climate projections, Journal of geophysical research: atmospheres, 118, 2473–2493, 2013.
- Steinert, N. J. and Sanderson, B.: benmsanderson/METEOR: v1.0.2 (v1.0.2), Zenodo, https://doi.org/zenodo.15727973, 2025.
  - Tebaldi, C. and Knutti, R.: The use of the multi-model ensemble in probabilistic climate projections, Philosophical Transactions of the Royal Society A, 365, 2053–2075, https://doi.org/10.1098/rsta.2007.2076, 2007.
  - Tebaldi, C., Debeire, K., Eyring, V., Fischer, E., Fyfe, J., Friedlingstein, P., Knutti, R., Lowe, J., O'Neill, B., Sanderson, B., van Vuuren, D., Riahi, K., Meinshausen, M., Nicholls, Z., Tokarska, K. B., Hurtt, G., Kriegler, E., Lamarque, J.-F., Meehl, G., Moss, R., Bauer, S. E.,
- Boucher, O., Brovkin, V., Byun, Y.-H., Dix, M., Gualdi, S., Guo, H., John, J. G., Kharin, S., Kim, Y., Koshiro, T., Ma, L., Olivié, D., Panickal, S., Qiao, F., Rong, X., Rosenbloom, N., Schupfner, M., Séférian, R., Sellar, A., Semmler, T., Shi, X., Song, Z., Steger, C., Stouffer, R., Swart, N., Tachiiri, K., Tang, Q., Tatebe, H., Voldoire, A., Volodin, E., Wyser, K., Xin, X., Yang, S., Yu, Y., and Ziehn, T.: Climate model projections from the Scenario Model Intercomparison Project (ScenarioMIP) of CMIP6, Earth System Dynamics, 12, 253–293, https://doi.org/10.5194/esd-12-253-2021, 2021.
- Tebaldi, C., Snyder, A., and Dorheim, K.: STITCHES: creating new scenarios of climate model output by stitching together pieces of existing simulations, Earth System Dynamics, 13, 1557–1609, https://doi.org/10.5194/esd-13-1557-2022, 2022.
  - van Vuuren, D. P., Bayer, L. B., Chuwah, C., Ganzeveld, L., Hazeleger, W., van den Hurka, B., van Noije, T., O'Neill, B., and Strengers, B. J.: A comprehensive view on climate change: coupling of earth system and integrated assessment models, Environmental Research Letters, 7, https://doi.org/10.1088/1748-9326/7/2/024012, 2012.
- Watson-Parris, D.: ClimateBench, https://doi.org/10.5281/zenodo.7064308, 2021.
  - Watson-Parris, D., Williams, A., Deaconu, L., and Stier, P.: Model calibration using ESEm v1.1.0 an open, scalable Earth system emulator, Geoscientific Model Development, 14, 7659–7672, https://doi.org/10.5194/gmd-14-7659-2021, 2021.
  - Watson-Parris, D., Rao, Y., Olivié, D., Seland, Ø., Nowack, P., Camps-Valls, G., Stier, P., Bouabid, S., Dewey, M., Fons, E., Gonzalez, J., Harder, P., Jeggle, K., Lenhardt, J., Manshausen, P., Novitasari, M., Ricard, L., and Roesch, C.: ClimateBench
- v1.0: A Benchmark for Data-Driven Climate Projections, Journal of Advances in Modeling Earth Systems, 14, e2021MS002954, https://doi.org/https://doi.org/10.1029/2021MS002954, e2021MS002954 2021MS002954, 2022.
  - Wilcox, L. J., Allen, R. J., Samset, B. H., Bollasina, M. A., Griffiths, P. T., Keeble, J., Lund, M. T., Makkonen, R., Merikanto, J., O'Donnell, D., Paynter, D. J., Persad, G. G., Rumbold, S. T., Takemura, T., Tsigaridis, K., Undorf, S., and Westervelt, D. M.: The Regional Aerosol

- Model Intercomparison Project (RAMIP), Geoscientific Model Development, 16, 4451–4479, https://doi.org/10.5194/gmd-16-4451-2023, 2023.
  - Womack, C. B., Giani, P., Eastham, S. D., and Selin, N. E.: Rapid Emulation of Spatially Resolved Temperature Response to Effective Radiative Forcing, Journal of Advances in Modeling Earth Systems, 17, e2024MS004523, https://doi.org/10.1029/2024MS004523, e2024MS004523 2024MS004523, 2025.
- Zelazowski, P., Huntingford, C., Mercado, L. M., and Schaller, N.: Climate pattern-scaling set for an ensemble of 22 GCMs adding uncertainty to the IMOGEN version 2.0 impact system, Geoscientific Model Development, 11, 541–560, https://doi.org/10.5194/gmd-11-541-2018, 2018.
  - Zhao, A. D., Stevenson, D. S., and Bollasina, M. A.: The role of anthropogenic aerosols in future precipitation extremes over the Asian Monsoon Region, Climate Dynamics, 52, 6257–6278, https://doi.org/10.1007/s00382-018-4514-7, 2019.
- Zhao, B., Reichler, T., Strong, C., and Penland, C.: Simultaneous Evolution of Gyre and Atlantic Meridional Overturning Circulation Anomalies as an Eigenmode of the North Atlantic System, Journal of Climate, 30, 6737 6755, https://doi.org/10.1175/JCLI-D-16-0751.1, 2017.