# Peer review of "METEORv1.0.1: A novel framework for emulating multi-timescale regional climate responses"

_EGUsphere, 2025_

## Referee Comment (RC2)

**Review for**

**METEORv1.0.1: A novel framework for emulating multi-timescale regional climate responses**

This manuscript describes the new spatial climate emulator METEOR. The model represents the spatial responses to greenhouse gas forcings and aerosols forcings separately using impulse response functions & patterns. The model itself is an improvement of classic pattern scaling approaches, thus filling in an important research gap. The method and the validation are overall good, albeit some minor flaws/suggestions. The manuscript is written clearly, especially its storyline. The figures are quite complete, and easy enough to understand quickly. I think that this manuscript would completely fit the scope of GMD, and could be published with minor revisions. I will describe below the suggested revisions.

**Minor flaws:**

My principal point would be on the validation. The current framework is (1) deconvolving the global signals, (2) validation with the Pearson coefficients (Table 1), (3) deducing spatial patterns. The validation is indeed very good globally, but there is for now no validation of the spatial ouputs of the whole modelling chain. To evaluate the spatial decomposition, I would appreciate a map of the R2 for in-sample and out-of-sample scenarios, for comparison of the original data to the emulated data.

L85-90 & L256-257: training responses on single experiments is somewhat of a risk, reducing the domain of validity of the emulator. For instance, although IRFs are reasonable approximations[1], IRFs for the response atmospheric fraction of CO2 to a pulse of CO2 emissions is known to depend on its calibration under preindustrial and current conditions[2]. This is mostly because the preindustrial carbon cycle does not behave exactly like a perturbed carbon cycle. Here, some IRFs of METEOR are calibrated with *abrupt-4xCO2*, thus starting under a preindustrial climate, up to a disturbed climate.

Have you tried training on multiple experiments instead, using both experiments under past and current conditions? For instance with the variants of *ssp245* as well for GhG, and some variants of *historical* with only aerosols?

Related question L96-100 and L181-199, in particular equation 9: Why use the difference when there is *ssp245-aer*? Besides *ssp245-GHG, ssp245-CO2, ssp245-stratO3*, etc? Quick insight, there may some differences if using different combinations of experiments. For instance, using *hist-aer* would lead to different results than using *historical – hist-PiAer* or *historical – hist-piNTCF*. Because the temporal response of a forcing may depend on the other forcings. The response under *hist-aer* has much less warning, different atmospheric chemistry for aerosols than what we would see under *historical – hist-PiAer*. Though, I agree that it is a second order

effect, tough to include in this framework. Thus I would simply suggest you to mention this limitation.

L101-110: I would be careful about summarizing aerosols with sulfates. Each aerosol would have its own specificities in terms of radiative effects, atmospheric chemistry, lifetimes and transport. Some experiments that would be useful here would be: *hist-aer, hist-piNTCF, hist-stratO3, hist-piAer*. I am not asking to recalibrate the model, that would represent a massive additional work to account for different aerosol species. But it could be noted as a potential limitation for future research.

L144-150 + 199-201: I appreciate the technique of separating the timescales into these bins, and great work to evaluate the adequate numbers of timescales in Appendix A. Though I have a question on the choice of bounds. For now, the minimum $\tau$ is 1 year. It assumes that there is a non-immediate stabilization of the response to the forcing, which is physically quite robust in this context. Though, it is not unlikely that there may be one mode for the response below 1 year, for a very rapid stabilization. Of course the model runs at an annual resolution, but it would give some flexibility to the response at t=1 and the asymptote (equation 2). Maybe even more relevant for aerosols? So my question would be: would there be a significant gain in performance by including another mode below 1 year?

By the way, looking at Table B1, there are lots of $\tau_1$ at 1.0 year, which could be a sign that the minimization algorithm forced the $\tau_1$ at the very limit of what it could do given the user-defined bounds, in other words that the error function could be minimized by relaxing these bounds. Overall, about half of the tau of the table seems to hit their bounds. I think that it should be investigated.

**Suggestions:**

L24: can add as well RCMIP phase 2, probably more relevant than RCMIP phase 1.

L26-27: the uses for fast spatial climate modelling frameworks is broader than that, see for instance[3-6]

L35-38: Important point on probabilistic spatial climate information & pattern scaling. Pattern scaling provides only a deterministic response, the mean of the climate field. Having a probabilistic information may come either from the uncertainty in modelling or through natural variability. Pure pattern scaling like PRIME does not include natural variability. MESMER does represent the natural variability obtained through temporal auto-regressions with spatially correlated innovations. Then, regarding precipitations, this is obtained through a more elaborated approach[7]. Finally, pattern scaling is generalized with non-stationary distributions with MESMER-X[8,9], while also removing the assumption of linearity, eg for soil moisture. For the

sake of transparency, I'm the author of the two latter papers, and I'm not asking the authors to add them as references.

My point is that there is a need for clarification, that pure pattern scaling is limited to uncertainties in modelling for probabilistic assessment, and going further with natural variability requires additional statistical tools.

L38-41: In STITCHES, the portions of existing simulations are found not only through the median in global mean temperature, but also by its derivative.

L56-58: The feedbacks of JULES don't feedback in FaIR or PRIME (yet).

L17-137: At resolution of the ESM? Is there any rescaling?

L140-143: decomposing $x_{global}$ with IRFs is a good idea, it simplifies the modelling[10]. Another approach is to decompose the local effects with IRFs, e.g. $T_{GHG}$, as conducted in Womack et al 2025[11]. In my opinion, both approaches have pros and cons, I'm not sure which one performs best, but Womack et al, 2025 should be mentioned.

L211: the link between the Moore-Penrose pseudoinverse and Barata and Hussen, 2012[12] should be clearer. In the text, the paper was mentioned without the method, and here the method is mentioned without the paper.

Figures 4 & 5: typos in *historical*, and later the the.

Figures 3 & 4: would you have any insights why the signal for polar amplification is so clear for aerosols (fig 4) but not that much for GhG (fig 3)?

1       Myhre, G. *et al*. PDRMIP: A Precipitation Driver and Response Model Intercomparison Project—Protocol and Preliminary Results. *Bulletin of the American Meteorological Society* **98**, 1185-1198 (2017). https://doi.org/https://doi.org/10.1175/BAMS-D-16-0019.1
2       Joos, F. *et al*. Carbon dioxide and climate impulse response functions for the computation of greenhouse gas metrics: A multi-model analysis. *Atmospheric Chemistry and Physics* **13**, 2793-2825 (2013).
3       van Vuuren, D. P. *et al*. A comprehensive view on climate change: coupling of earth system and integrated assessment models. *Environmental Research Letters* **7**, 024012 (2012). https://doi.org/10.1088/1748-9326/7/2/024012

4      Kikstra, J. S. *et al.* The social cost of carbon dioxide under climate-economy feedbacks and temperature variability. *Environmental Research Letters* **16**, 094037 (2021). https://doi.org/10.1088/1748-9326/ac1d0b

5      Dietz, S., Rising, J., Stoerk, T. & Wagner, G. Economic impacts of tipping points in the climate system. *Proceedings of the National Academy of Sciences* **118**, e2103081118 (2021). https://doi.org/10.1073/pnas.2103081118

6      Rising, J., Tedesco, M., Piontek, F. & Stainforth, D. A. The missing risks of climate change. *Nature* **610**, 643-651 (2022). https://doi.org/10.1038/s41586-022-05243-6

7      Schöngart, S. *et al.* Introducing the MESMER-M-TPv0.1.0 module: Spatially Explicit Earth System Model Emulation for Monthly Precipitation and Temperature. *EGUsphere* **2024**, 1-51 (2024). https://doi.org/10.5194/egusphere-2024-278

8      Quilcaille, Y., Gudmundsson, L., Beusch, L., Hauser, M. & Seneviratne, S. I. Showcasing MESMER-X: Spatially Resolved Emulation of Annual Maximum Temperatures of Earth System Models. *Geophysical Research Letters* **49**, e2022GL099012 (2022). https://doi.org/10.1029/2022GL099012

9      Quilcaille, Y., Gudmundsson, L. & Seneviratne, S. I. Extending MESMER-X: a spatially resolved Earth system model emulator for fire weather and soil moisture. *Earth Syst. Dynam.* **14**, 1333-1362 (2023). https://doi.org/10.5194/esd-14-1333-2023

10     Proistosescu, C. & Huybers, P. J. Slow climate mode reconciles historical and model-based estimates of climate sensitivity. *Science Advances* **3**, e1602821 (2017). https://doi.org/10.1126/sciadv.1602821

11     Womack, C. B., Giani, P., Eastham, S. D. & Selin, N. E. Rapid Emulation of Spatially Resolved Temperature Response to Effective Radiative Forcing. *Journal of Advances in Modeling Earth Systems* **17**, e2024MS004523 (2025). https://doi.org/https://doi.org/10.1029/2024MS004523

12     Barata, J. C. A. & Hussein, M. S. The Moore–Penrose Pseudoinverse: A Tutorial Review of the Theory. *Brazilian Journal of Physics* **42**, 146-165 (2012). https://doi.org/10.1007/s13538-011-0052-z

---

## Author Response (AR1)

**Reply to reviews on: "METEORv1.0.1: A novel framework for emulating multi-timescale regional climate responses"**

**Reviewer 1**

**Review of METEOR 1.0**

**Summary of paper:** The authors show a new pattern-scaling technique in which regional annual-mean temperature and precipitation patterns, as responses to forcing changes, are time-dependent. The results capture the multi-model mean of assessed CMIP6 responses quite well, with an RMSE of  $\sim 0.15$  K for warming and  $0.16 \times 10^{-7}$  kg m-2 s-1 for temperature.

**Recommendation:** Acceptance after revisions (whether those are minor or major is in the eye of the beholder and up to the authors).

**General comments:**

A very useful contribution to the long quest to emulate GCM/ESM response fields via enhanced pattern-scaling techniques.

**CMIP6 MMM versus CMIP6 individual-model emulator:**

As currently presented, the paper features mainly the validation of METEOR against the CMIP6 MMM, rather than validations of individual CMIP6 models. This is not made clear in the abstract or most of the text, where the reader gets the impression that METEOR, in its current calibration, is a useful emulator of individual ESMs. That might be the case, but it is not shown. In other words, the paper is not clearly framed as being limited to emulating only the multi-model mean. If the authors wish to present METEOR as an individual-GCM/ESM emulator, then the paper needs to test the appropriateness of CMIP6 model-by-model responses. Only small in-sample goodness-of-fit metrics are shown (e.g., Figure A1 panels a and d present the RMSE for the GHG response in the in-sample abrupt-4×CO2). I therefore strongly encourage the authors to show more model-by-model validation—for example, by including absolute-error maps of 20-year means for individual model SSP5-8.5 or SSP1-2.6 out-of-sample temperature and precipitation fields for 2080–2100. Tables of RMSE and MAE values by model and scenario would be useful in an Appendix, allowing comparison to alternative emulation techniques. Similarly, Figures 8 and 9 could be

extended to include maps of the best and worst CMIP6 model fits, rather than showing only MMM differences.

**Global-mean validation versus regional validation:**

At present, Figures 5–7 and B13–B20 show useful comparison plots for global-mean temperature and precipitation responses. That is reassuring (and a great result), but for an emulator of regional climate responses, more regional comparisons are needed. The global-mean response can be obtained much more simply—e.g., as an extension of the C-SCM with a few lines of code and these calibration parameters. I suggest replacing (or extending) Figures B13–B20 with figures that show the worst- and best-performing regions (using either custom definitions or IPCC AR6 regions). Regional responses could also be shown as maps—you already include CMIP6 MMM comparison maps in your Figures 8 and 9.

**Limitations for impact models:**

The utility of these results for impact emulators depends on each emulator's needs. METEOR v1.0 is limited to annual-mean projections of best-estimate warming and precipitation changes, and does not yet include variability, compound-event modeling, climate-oscillation modes, distribution tails, etc. Although some of these caveats are mentioned in the conclusion, an explicit upfront statement of the current emulator's scope (and its limitations) would be helpful.

**Physical interpretation of response patterns (Figures B1–B12):**

Looking at the GHG and "residual" response patterns, one wonders whether they are intended purely as statistical fits (in which case they need not be physically interpretable, as long as applications stay within the training spectrum), or whether they represent physically meaningful patterns. If the latter, one could apply the emulator beyond 2100 to 2300 with more confidence. Since the authors do not clearly state that these are statistical fits—and some discussion refers to physical interpretation of short- and long-term responses—I suggest the following:

1. **Equilibrium response aggregate pattern:** Add a fourth column to Figures 3 and 4, as well as B1–B12, that sums the short-, medium-, and long-term response patterns. This should yield the equilibrium response pattern, which readers can then evaluate for physical plausibility. If the equilibrium response is not physically plausible (and some of the patterns seem hard to interpret), then these components should be framed explicitly as purely statistical fits valid up to 2100 for the shown

validations. Alternatively, you might introduce training constraints—for example, requiring that the sum of the three timescales falls within a physically plausible range. You could also discuss whether the land-ocean warming ratio evolves plausibly from short-term through equilibrium response.

2. **Full colorbar:** Many patterns appear clipped by the chosen colorbar limits, making it hard to see true minima and maxima. Please include a full colorbar for these figures and choose its range to include extreme values (possibly on a logarithmic scale) so that readers can see tail-end values. For example, in Figure B6 the MIROC-ES2L long-term GHG precipitation response is unclear; likewise CanESM5's short-term precipitation response in Figure B5 and UKESM1-0-LL's medium-term temperature response in Figure B3.

**Correlation between temperature and precipitation:**

Since METEOR emulates both variables, it would be useful to examine their regional coevolution. For instance, map percent precipitation change per degree of warming—some regions should show  $\sim 2-5$  % °C-1, moisture-saturated regions near Clausius–Clapeyron ( $\sim 7$  % °C-1), etc. This would provide a physics-based check on the emulator's joint behavior.

**Skill comparison to other techniques:**

The reported skill metrics (Pearson, RMSE) need context. Consider benchmarking against the ClimateBench test (doi:10.1029/2021MS002954) using NorESM2 output, or comparing to other published emulators. You might also compare each model's emulation error to the inter-model spread in response patterns, to assess whether emulator errors are small relative to GCM diversity.

**Small comments:**

- Lines 11–12, Abstract: You state that the emulation system can "accurately predict gridded responses to out-of-sample scenarios." That is too broad, since you demonstrate accuracy only for the MMM, annual means, and expected values. Please qualify.
- **Line 47:** Do you mean that ClimateBench data are not widely available? They are provided via Zenodo—please clarify.
- **Line 105:** "most impactful non-GHG forcer." Perhaps note that this is currently true but may differ under low-emission scenarios.

- **Line 134:** When you subtract the piControl "climatology," do you mean a 20- or 30-year rolling mean, a trend, or a non-parametric low-pass filter? Please specify.
- **Line 137:** Clarify whether you use cos(lat) for area weighting or each model's native areacella.
- **Figures 3 & 4:** Much of the long-term precipitation response lies outside the colorbar range—consider widening it or otherwise showing pattern extrema.
- Figure A1 caption: Typo: "fo" → "of."
- **Tropospheric ozone response:** Where is this captured? I assume in the residual (aerosol-scaled) response—please state.
- Residual scaling bias: Using sulfate as the scaler for residual response may bias low-emission scenarios, since nitrate aerosols could dominate forcing by century's end. Discuss this potential bias.

**Reviewer 2**

Review for METEORv1.0.1: A novel framework for emulating multitimescale regional climate responses

This manuscript describes the new spatial climate emulator METEOR. The model represents the spatial responses to greenhouse gas forcings and aerosols forcings separately using impulse response functions & patterns. The model itself is an improvement of classic pattern scaling approaches, thus filling in an important research gap. The method and the validation are overall good, albeit some minor flaws/suggestions. The manuscript is written clearly, especially its storyline. The figures are quite complete, and easy enough to understand quickly. I think that this manuscript would completely fit the scope of GMD, and could be published with minor revisions. I will describe below the suggested revisions.

**Minor flaws:**

My principal point would be on the validation. The current framework is (1) deconvolving the global signals, (2) validation with the Pearson coefficients (Table 1), (3) deducing spatial patterns. The validation is indeed very good globally, but there is for now no validation of the spatial ouputs of the whole modelling chain. To evaluate the spatial decomposition, I would appreciate a map of the R2 for in-sample and out-of-sample scenarios, for comparison of the original data to the emulated data.

L85-90 & L256-257: training responses on single experiments is somewhat of a risk, reducing the domain of validity of the emulator. For instance, although IRFs are reasonable

approximations 1, IRFs for the response atmospheric fraction of CO2 to a pulse of CO2 emissions is known to depend on its calibration under preindustrial and current conditions 2. This is mostly because the preindustrial carbon cycle does not behave exactly like a perturbed carbon cycle. Here, some IRFs of METEOR are calibrated with abrupt-4xCO2, thus starting under a preindustrial climate, up to a disturbed climate. Have you tried training on multiple experiments instead, using both experiments under past and current conditions? For instance with the variants of ssp245 as well for GhG, and some variants of historical with only aerosols?

Related question L96-100 and L181-199, in particular equation 9: Why use the difference when there is ssp245-aer? Besides ssp245-GHG, ssp245-CO2, ssp245-stratO3, etc? Quick insight, there may some differences if using different combinations of experiments. For instance, using hist-aer would lead to different results than using historical – hist-PiAer or historical – histpiNTCF. Because the temporal response of a forcing may depend on the other forcings. The response under hist-aer has much less warning, different atmospheric chemistry for aerosols than what we would see under historical – hist-PiAer. Though, I agree that it is a second order effect, tough to include in this framework. Thus I would simply suggest you to mention this limitation.

L101-110: I would be careful about summarizing aerosols with sulfates. Each aerosol would have its own specificities in terms of radiative effects, atmospheric chemistry, lifetimes and transport. Some experiments that would be useful here would be: hist-aer, hist-piNTCF, histstratO3, hist-piAer. I am not asking to recalibrate the model, that would represent a massive additional work to account for different aerosol species. But it could be noted as a potential limitation for future research.

L144-150 + 199-201: I appreciate the technique of separating the timescales into these bins, and great work to evaluate the adequate numbers of timescales in Appendix A. Though I have a question on the choice of bounds. For now, the minimum  $\tau$  is 1 year. It assumes that there is a non-immediate stabilization of the response to the forcing, which is physically quite robust in this context. Though, it is not unlikely that there may be one mode for the response below 1 year, for a very rapid stabilization. Of course the model runs at an annual resolution, but it would give some flexibility to the response at t=1 and the asymptote (equation 2). Maybe even more relevant for aerosols? So my question would be: would there be a significant gain in performance by including another mode below 1 year? By the way, looking at Table B1, there are lots of  $\tau$ 1 at 1.0 year, which could be a sign that the minimization algorithm forced the  $\tau$ 1 at the very limit of what it could do given the user-defined bounds, in other words that the error function could be minimized by relaxing these

bounds. Overall, about half of the tau of the table seems to hit their bounds. I think that it should be investigated.

Suggestions: L24: can add as well RCMIP phase 2, probably more relevant than RCMIP phase 1.

L26-27: the uses for fast spatial climate modelling frameworks is broader than that, see for instance3-6

L35-38: Important point on probabilistic spatial climate information & pattern scaling. Pattern scaling provides only a deterministic response, the mean of the climate field. Having a probabilistic information may come either from the uncertainty in modelling or through natural variability. Pure pattern scaling like PRIME does not include natural variability. MESMER does represent the natural variability obtained through temporal autoregressions with spatially correlated innovations. Then, regarding precipitations, this is obtained through a more elaborated approach7. Finally, pattern scaling is generalized with non-stationary distributions with MESMER-X 8,9, while also removing the assumption of linearity, eg for soil moisture. For the sake of transparency, I'm the author of the two latter papers, and I'm not asking the authors to add them as references. My point is that there is a need for clarification, that pure pattern scaling is limited to uncertainties in modelling for probabilistic assessment, and going further with natural variability requires additional statistical tools.

L38-41: In STITCHES, the portions of existing simulations are found not only through the median in global mean temperature, but also by its derivative.

L56-58: The feedbacks of JULES don't feedback in FaIR or PRIME (yet).

L17-137: At resolution of the ESM? Is there any rescaling?

L140-143: decomposing xglobal with IRFs is a good idea, it simplifies the modelling 10. Another approach is to decompose the local effects with IRFs, e.g. TGHG, as conducted in Womack et al 2025 11. In my opinion, both approaches have pros and cons, I'm not sure which one performs best, but Womack et al, 2025 should be mentioned.

L211: the link between the Moore-Penrose pseudoinverse and Barata and Hussen, 201212 should be clearer. In the text, the paper was mentioned without the method, and here the method is mentioned without the paper.

Figures 4 & 5: typos in historical, and later the the.

Figures 3 & 4: would you have any insights why the signal for polar amplification is so clear for aerosols (fig 4) but not that much for GhG (fig 3)?

- 1 Myhre, G. et al. PDRMIP: A Precipitation Driver and Response Model Intercomparison Project—Protocol and Preliminary Results. Bulletin of the American Meteorological Society 98, 1185-1198 (2017). https://doi.org/https://doi.org/10.1175/BAMS-D-16-0019.1
- 2 Joos, F. et al. Carbon dioxide and climate impulse response functions for the computation of greenhouse gas metrics: A multi-model analysis. Atmospheric Chemistry and Physics 13, 2793-2825 (2013).
- 3 van Vuuren, D. P. et al. A comprehensive view on climate change: coupling of earth system and integrated assessment models. Environmental Research Letters 7, 024012 (2012). https://doi.org/10.1088/1748-9326/7/2/024012
- 4 Kikstra, J. S. et al. The social cost of carbon dioxide under climate-economy feedbacks and temperature variability. Environmental Research Letters 16, 094037 (2021). https://doi.org/10.1088/1748-9326/ac1d0b
- 5 Dietz, S., Rising, J., Stoerk, T. & Wagner, G. Economic impacts of tipping points in the climate system. Proceedings of the National Academy of Sciences 118, e2103081118 (2021). https://doi.org/10.1073/pnas.2103081118
- 6 Rising, J., Tedesco, M., Piontek, F. & Stainforth, D. A. The missing risks of climate change. Nature 610, 643-651 (2022). https://doi.org/10.1038/s41586-022-05243-6
- 7 Schöngart, S. et al. Introducing the MESMER-M-TPv0.1.0 module: Spatially Explicit Earth System Model Emulation for Monthly Precipitation and Temperature. EGUsphere 2024, 1-51 (2024). https://doi.org/10.5194/egusphere-2024-278
- 8 Quilcaille, Y., Gudmundsson, L., Beusch, L., Hauser, M. & Seneviratne, S. I. Showcasing MESMER-X: Spatially Resolved Emulation of Annual Maximum Temperatures of Earth System Models. Geophysical Research Letters 49, e2022GL099012 (2022). https://doi.org/10.1029/2022GL099012
- 9 Quilcaille, Y., Gudmundsson, L. & Seneviratne, S. I. Extending MESMER-X: a spatially resolved Earth system model emulator for fire weather and soil moisture. Earth Syst. Dynam. 14, 1333-1362 (2023). https://doi.org/10.5194/esd-14-1333-2023 10 Proistosescu, C. & Huybers, P. J. Slow climate mode reconciles historical and modelbased estimates of climate sensitivity. Science Advances 3, e1602821 (2017). https://doi.org/10.1126/sciadv.1602821
- 11 Womack, C. B., Giani, P., Eastham, S. D. & Selin, N. E. Rapid Emulation of Spatially Resolved Temperature Response to Effective Radiative Forcing. Journal of Advances in Modeling Earth Systems 17, e2024MS004523 (2025).

https://doi.org/https://doi.org/10.1029/2024MS004523

12 Barata, J. C. A. & Hussein, M. S. The Moore–Penrose Pseudoinverse: A Tutorial Review of the Theory. Brazilian Journal of Physics 42, 146-165 (2012). https://doi.org/10.1007/s13538-011-0052-z

**Reply to reviewer 1**

Thank you for your thorough read, and useful and constructive comments. We have considered them all and made adjustments, clarifications or additions accordingly, all of which we feel strengthen the manuscript. Below we answer the concrete comments one by one in greater detail.

"As currently presented, the paper features mainly the validation of METEOR against the CMIP6 MMM, rather than validations of individual CMIP6 models. This is not made clear in the abstract or most of the text, where the reader gets the impression that METEOR, in its current calibration, is a useful emulator of individual ESMs. That might be the case, but it is not shown. In other words, the paper is not clearly framed as being limited to emulating only the multi-model mean. If the authors wish to present METEOR as an individual-GCM/ESM emulator, then the paper needs to test the appropriateness of CMIP6 model-bymodel responses. Only small in-sample goodness-of-fit metrics are shown (e.g., Figure A1 panels a and d present the RMSE for the GHG response in the in-sample abrupt-4×CO2). I therefore strongly encourage the authors to show more model-by-model validation—for example, by including absolute-error maps of 20-year means for individual model SSP5-8.5 or SSP1-2.6 out-of-sample temperature and precipitation fields for 2080–2100. Tables of RMSE and MAE values by model and scenario would be useful in an Appendix, allowing comparison to alternative emulation techniques. Similarly, Figures 8 and 9 could be extended to include maps of the best and worst CMIP6 model fits, rather than showing only MMM differences."

We agree that the framing of the figures is skewed towards the multi-model mean, but we do mean to convey the applicability of METEOR as an individual model emulator. The appendix has many individual model plots, and we will update these and make the individual model results clearer. We will also upload spatial fit plots for all the models that we ran METEOR with in a Zenodo repository. However, the reason why we focus more on the multi-model mean, is that the choices of models would be somewhat arbitrary, and there are really a lot of models to include. Defending the choice of one particular model over the other can then be tricky. However, noting your comment, we have chosen now to include plots for all the models for which we had data for ssp5-3.4-over in plots individually in the main text, as this was a limited number of models, illustrating fits for an out of

sample scenario, and they include overshoot - the modelling of which is a particular motivation for METEOR. We fully agree with the reviewer that error metric tables for all models and scenarios are useful in the appendix and have added them accordingly.

"At present, Figures 5–7 and B13–B20 show useful comparison plots for global-mean temperature and precipitation responses. That is reassuring (and a great result), but for an emulator of regional climate responses, more regional comparisons are needed. The global-mean response can be obtained much more simply—e.g., as an extension of the C-SCM with a few lines of code and these calibration parameters. I suggest replacing (or extending) Figures B13–B20 with figures that show the worst- and best-performing regions (using either custom definitions or IPCC AR6 regions). Regional responses could also be shown as maps—you already include CMIP6 MMM comparison maps in your Figures 8 and 9."

We agree with this criticism and hope that the addition of the new main-text figures is useful in that regard. They show regional model versus emulation scatter plots per region for 9 different regions, and regionally separated plots of change in precipitation versus temperature change per model for both direct output and emulation in the over-shoot scenario for each of 8 regions, to both show better how well the model can fit both per model and per region. For ssp5-3.4-over, we also show maps of fits per model in the appendix. We also hope that the above-mentioned spatial maps for every model uploaded to a Zenodo repository will help with this.

"The utility of these results for impact emulators depends on each emulator's needs.

METEOR v1.0 is limited to annual-mean projections of best-estimate warming and precipitation changes, and does not yet include variability, compound-event modeling, climate-oscillation modes, distribution tails, etc. Although some of these caveats are mentioned in the conclusion, an explicit upfront statement of the current emulator's scope (and its limitations) would be helpful."

We will make the limitations of METEOR v1.0 clearer also earlier. We would like to point out though that, although it has not been validated outside of the mean annual variables considered in the paper, the setup is a bit less restrictive than this, as METEOR can in principle model other annual-mean projected variables for which there is data, and for which the underlaying assumption of forcing driven timescale patterns holds, that of course does not include any of the implications you list here.

"Looking at the GHG and "residual" response patterns, one wonders whether they are intended purely as statistical fits (in which case they need not be physically interpretable, as long as applications stay within the training spectrum), or whether they represent physically meaningful patterns. If the latter, one could apply the emulator beyond 2100 to 2300 with more confidence. Since the authors do not clearly state that these are statistical fits—and some discussion refers to physical interpretation of short- and long-term responses—I suggest the following:

- 1. **Equilibrium response aggregate pattern:** Add a fourth column to Figures 3 and 4, as well as B1–B12, that sums the short-, medium-, and long-term response patterns. This should yield the equilibrium response pattern, which readers can then evaluate for physical plausibility. If the equilibrium response is not physically plausible (and some of the patterns seem hard to interpret), then these components should be framed explicitly as purely statistical fits valid up to 2100 for the shown validations. Alternatively, you might introduce training constraints—for example, requiring that the sum of the three timescales falls within a physically plausible range. You could also discuss whether the land-ocean warming ratio evolves plausibly from short-term through equilibrium response.
- 2. Full colorbar: Many patterns appear clipped by the chosen colorbar limits, making it hard to see true minima and maxima. Please include a full colorbar for these figures and choose its range to include extreme values (possibly on a logarithmic scale) so that readers can see tail-end values. For example, in Figure B6 the MIROC-ES2L long-term GHG precipitation response is unclear; likewise CanESM5's short-term precipitation response in Figure B5 and UKESM1-0-LL's medium-term temperature response in Figure B3.

,,

We thank the reviewer for this and agree that some more discussion on the interpretability is in order. The model does in principle only provide statistical fits, but they should contain physical information, the first point here also has led us to reevaluate and redo our figures 3, 4 and B1-B16. Though it is in principle true that the sum of the three patterns is the equilibrium response of the model, this is not really a meaningfully constrained quantity from the model and nor is the display of the three patterns in the way shown in these figures. For the shortest timescale, the equilibrium pattern response makes sense, but as the timescales increase, there is an ambiguity between the amplitude of the response and the timescale of equilibration – such that extrapolating beyond the timescale of the training data to an equilibration in hundreds or thousands of years is highly under-constrained.

The fit and training data fit less and less to the equilibrium response, and more and more to the linear or only the few first terms in the Taylor expansion of equations 3 and 10 for the time. In the equilibrium case the fit would be fitting the underlaying pattern, BGHG and Baer, but with only 150 years of simulations (or less), the fit for the longest timescale only ever has training data to fit something like 1/tau\_K \*BGHG (i.e. linear regime), so the relative strength of the pattern is dampened significantly. As the training data does not go over this, we also do not expect the model to yield reliable results for runs that are significantly prolonged in time. I.e. we have much more trust in out of sample results for different forcing/emissions-pathways than in the extensions of the runs to very much longer timescales (for that longer training datasets would be needed). In effect, the time-evolving response of the slowest modes to a step change in forcing appears to be a straight line in the ~100 year training data considered here – there is simply no information on how that mode will equilibrate. For applications on the ~100 year timescale, this is not a problem – but the model is not suited to an extrapolation to equilibrium.

Exactly where the trustworthiness of the model ends is a topic which should be explored in the future, and we will mention this. What this also means is that the pattern comparison in these figures as they stand are not very meaningful for two reasons: 1. The three patterns are not scaled in a meaningful way so the relative strengths between them are not really reflective of the relative strengths between them in any part of the model which has any validity, and 2. The mean between models also doesn't make much sense here, as models with larger tau\_k values will have larger relative weight, particularly for the long timescale patterns, making the summation and mean between them not particularly instructive, and hence the physical interpretation of them even less so. To amend this, we have now reframed the plots to show the contribution of different timescales to the total warming response 100 years after an abrupt change. We think this is a more meaningful illustration, given it shows the relatively minor contribution of the slow timescales to the total response in year 100 – but does not imply any confidence in extrapolation to longer time scales

This is achieved by scaling the patterns to their mean value in years 80-120 in the reconstruction of abrupt-4xCO2 for the GHG patterns and 1980-2020 in the reconstruction of historical-ssp245 for the aerosol patterns. This way we can also add them together to produce a meaningful summed pattern, and the multi-model-mean will be fairly weighted between models, showing results within the valid range for the emulator. These updates also solve the colorbar issue as the very large amplitude of the longest timescale pattern is appropriately dampened by its temporal coefficient.

"Since METEOR emulates both variables, it would be useful to examine their regional coevolution. For instance, map percent precipitation change per degree of warming—some regions should show ~2–5 % °C-1, moisture-saturated regions near Clausius–Clapeyron (~7 % °C-1), etc. This would provide a physics-based check on the emulator's joint behavior."

We thank the reviewer for this suggestion and hope the new ssp5-3.4-over plots that show the relationship between temperature and precipitation change per model and model emulation globally and for 8 different regions can show the degree to which METEOR is able to capture the joint behaviour.

"The reported skill metrics (Pearson, RMSE) need context. Consider benchmarking against the ClimateBench test (doi:10.1029/2021MS002954) using NorESM2 output, or comparing to other published emulators. You might also compare each model's emulation error to the inter-model spread in response patterns, to assess whether emulator errors are small relative to GCM diversity."

We agree with this assessment and have therefore added a comparison to the ClimateBench test. We also include similar per model results in a new supplementary table. The skill metrics previously included are also comparable to skill metrics provided by the PRIME emulator, which we have pointed out in the text. We have also made spatial RMSE map plots for all scenarios.

**"Lines 11–12, Abstract:** You state that the emulation system can "accurately predict gridded responses to out-of-sample scenarios." That is too broad, since you demonstrate accuracy only for the MMM, annual means, and expected values. Please qualify."

We have now qualified this statement, however, we believe that with updated figures we also demonstrate accuracy at model level and for gridded responses, so the qualification is not as strong as suggested in this comment.

**"Line 47:** Do you mean that ClimateBench data are not widely available? They are provided via Zenodo—please clarify."

We understand that the statement could be misunderstood. What we mean here is that the ESM output available to train on for ClimateBench is not available for emulation across the majority of CMIP6 (and upcoming CMIP7) models. In this paper, we chose to train exclusively on output experiments which have CMIP6 outputs for all ESMs. We agree that some of the experiments used in ClimateBench could possibly yield better and more physically informative fits to aerosol forcing specifically, even for METEOR, however, the point here is that we think a setup that can be run for *any* CMIP6 ESM model is an advantage, and showcasing the model and it's performance on this dataset is therefore our priority. Both the *ssp370-lowNTCF* and the DAMIP experiments have been run by

considerably fewer of the ESMs. We have clarified this now in the text. METEOR can be run from lightly processed widely available CMIP6 data, and also comes with a zarrstore data download capability, which can be used to directly download and process the CMIP6 data available there, with the user not having to figure out any downloading or processing of CMIP data. We feel this greatly adds to the model's usability.

Producing an independent calibration of METEOR for a subset of models with a larger array of simulations (such as from climatebench) would certainly be valuable, but would imply a significantly different calibration pipeline. We feel this is beyond the scope of the current study.

"Line 105: "most impactful non-GHG forcer." Perhaps note that this is currently true but may differ under low-emission scenarios."

We agree with this point and have added a caveat in the text accordingly.

**"Line 134:** When you subtract the piControl "climatology," do you mean a 20- or 30-year rolling mean, a trend, or a non-parametric low-pass filter? Please specify."

We are subtracting the mean of the of the full piControl annual mean timeseries. We have added specification for this in the text.

**"Line 137:** Clarify whether you use cos(lat) for area weighting or each model's native areacella."

We use cos(lat), this is now specified.

**"Figures 3 & 4:** Much of the long-term precipitation response lies outside the colorbar range—consider widening it or otherwise showing pattern extrema."

The updated figure versions don't have this issue anymore.

"Tropospheric ozone response: Where is this captured? I assume in the residual (aerosol-scaled) response—please state."

We have not been clear enough on this point, but the aerosol-scaled residual response is exclusively mapped to sulphate aerosol related forcing. In our current setup, sulphate aerosol scales both direct sulphate aerosol forcing, and the totality of the aerosol-cloud interaction, but all other forcing responses, including tropospheric and stratospheric ozone, BC- and OC- aerosol direct forcing, and stratospheric water vapour forcing are mapped using the GHG-response patterns and timescale. Of course, as we are identifying the sulphate aerosol responses with a residual signal, any and all other forcers (and other uncertainties or inaccuracies in the GHG response modelling), will also feed into these, but

they are not directly included. We have tried to make this clearer in the text now, by adding a sentence to the Methods introduction section stating this.

"Residual scaling bias: Using sulfate as the scaler for residual response may bias lowemission scenarios, since nitrate aerosols could dominate forcing by century's end. Discuss this potential bias."

We have added some discussion on this point. Nitrate aerosol bias is, however, a wider problem as the forcing from nitrate aerosol is both not well-constrained and highly dependent on both emission location, height and sector.

**Reply to reviewer 2**

We thank the reviewer for constructive and careful reading of our manuscript, providing useful suggestions for improvement. Below we answer each point in detail and relay how we have attempted to improve the manuscript according to each of the criticisms.

"My principal point would be on the validation. The current framework is (1) deconvolving the global signals, (2) validation with the Pearson coefficients (Table 1), (3) deducing spatial patterns. The validation is indeed very good globally, but there is for now no validation of the spatial ouputs of the whole modelling chain. To evaluate the spatial decomposition, I would appreciate a map of the R2 for in-sample and out-of-sample scenarios, for comparison of the original data to the emulated data."

This comment aligns with comments from reviewer 1, and to improve this we have made the suggested maps, and in addition we have made a table of spatial and global errors comparing them to the ClimateBench test set and including the same error metrics for each model and scenario combination.

"L85-90 & L256-257: training responses on single experiments is somewhat of a risk, reducing the domain of validity of the emulator. For instance, although IRFs are reasonable approximations1, IRFs for the response atmospheric fraction of CO2 to a pulse of CO2 emissions is known to depend on its calibration under preindustrial and current conditions2. This is mostly because the preindustrial carbon cycle does not behave exactly like a perturbed carbon cycle. Here, some IRFs of METEOR are calibrated with abrupt-4xCO2, thus starting under a preindustrial climate, up to a disturbed climate. Have you tried training on multiple experiments instead, using both experiments under past and current conditions? For instance with the variants of ssp245 as well for GHG, and some variants of historical with only aerosols?"

While we agree that the IRF response assumption is a possible structural weakness of our approach, we also note that the Joos et al 2013 reference (2) also found that the "responses on temperature, sea level and ocean heat content is less sensitive to" CO2 concentration conditions. We note also that the same IRF assumption is made in simple climate models such as FaIR – and so, while it remains an approximation, it is a common and well understood approximation.

In METEOR, the emissions-to-forcing module (provided by CICERO-SCM) also allows for some state-sensitivity and dynamical response to previously emitted carbon, meaning that the forcing strength emulated is also affected by the background conditions to some degree. We have added a small paragraph to clarify this part of the model a bit further. As for training on more than one scenario, our current setup is done using a single experiment as training input per forcer, and more sophisticated combinations for training would require substantial changes in the training logic of METEOR. We therefore consider it out of scope for this article and leave such experimentation for future work.

"Related question L96-100 and L181-199, in particular equation 9: Why use the difference when there is ssp245-aer? Besides ssp245-GHG, ssp245-CO2, ssp245-stratO3, etc? Quick insight, there may some differences if using different combinations of experiments. For instance, using hist-aer would lead to different results than using historical – hist-PiAer or historical – histpiNTCF. Because the temporal response of a forcing may depend on the other forcings. The response under hist-aer has much less warning, different atmospheric chemistry for aerosols than what we would see under historical – hist-PiAer. Though, I agree that it is a second order effect, tough to include in this framework. Thus I would simply suggest you to mention this limitation."

We understand the suggestions made here, and in part we agree with them. In fact, in our first versions of the model, we used PDRMIP experiments (1) to train the model to fit multiple forcers one-by-one. One reason why we left this approach, for the use of residual fits, was to be able to use widely available and reasonably up-to-date model data from the CMIP6 ensemble, and to be able to emulate as wide a set of models as possible. For this we wanted to use only experiments which had been widely run for our emulation, at least for this demonstration of our model, choosing versatility and usefulness over the possibility of marginally higher accuracy. However, modelling using more specified experiments such as these to split into more forcer components, possibly even with regional split, is something we hope to do in the future and we touch on this point in the outlook section of the article.

"L101-110: I would be careful about summarizing aerosols with sulfates. Each aerosol would have its own specificities in terms of radiative effects, atmospheric chemistry,

lifetimes and transport. Some experiments that would be useful here would be: hist-aer, hist-piNTCF, histstratO3, hist-piAer. I am not asking to recalibrate the model, that would represent a massive additional work to account for different aerosol species. But it could be noted as a potential limitation for future research."

This point has now been clarified in our manuscript, we are in fact not conflating all aerosols with sulfate in the modelling. Instead, what we call aerosol patterns are only driven by and mapped using sulfate forcing (which also includes all aerosol-cloud interactions as they are sulfate driven in ciceroscm). All other forcing terms are mapped using the GHG-forcing responses. Also, the ciceroscm emissions-to-forcing modelling used as part of driving METEOR does, though in extremely simplified ways, account for lifetimes, radiative effects and some atmospheric chemistry on a per forcer basis for both aerosols and other individual forcer components. We have added a sentence to the text here to further clarify how we map the different forcing patterns.

"L144-150 + 199-201: I appreciate the technique of separating the timescales into these bins, and great work to evaluate the adequate numbers of timescales in Appendix A. Though I have a question on the choice of bounds. For now, the minimum  $\tau$  is 1 year. It assumes that there is a non-immediate stabilization of the response to the forcing, which is physically quite robust in this context. Though, it is not unlikely that there may be one mode for the response below 1 year, for a very rapid stabilization. Of course, the model runs at an annual resolution, but it would give some flexibility to the response at t=1 and the asymptote (equation 2). Maybe even more relevant for aerosols? So my question would be: would there be a significant gain in performance by including another mode below 1 year? By the way, looking at Table B1, there are lots of  $\tau$ 1 at 1.0 year, which could be a sign that the minimization algorithm forced the  $\tau$ 1 at the very limit of what it could do given the user-defined bounds, in other words that the error function could be minimized by relaxing these bounds. Overall, about half of the tau of the table seems to hit their bounds. I think that it should be investigated."

This point is valid, and we have done testing of the model using both overlapping timescales and shorter timescales than this. The solution presented in this paper was the best trade-off in terms of interpretability and performance. Given that the data and modelling are limited to annual resolution, the overall validity and interpretability of sub-yearly timescales would be somewhat questionable. We have now expanded the discussion a bit on this point to clarify why we have chosen this setup.

"Suggestions: L24: can add as well RCMIP phase 2, probably more relevant than RCMIP phase 1."

Thank you for the suggestion, we have added this reference.

"L26-27: the uses for fast spatial climate modelling frameworks is broader than that, see for instance3-6"

Thank you for the suggested widening of scope and proposed references which we have now included.

"L35-38: Important point on probabilistic spatial climate information & pattern scaling. Pattern scaling provides only a deterministic response, the mean of the climate field. Having a probabilistic information may come either from the uncertainty in modelling or through natural variability. Pure pattern scaling like PRIME does not include natural variability. MESMER does represent the natural variability obtained through temporal autoregressions with spatially correlated innovations. Then, regarding precipitations, this is obtained through a more elaborated approach? Finally, pattern scaling is generalized with non-stationary distributions with MESMER-X 8,9, while also removing the assumption of linearity, eg for soil moisture. For the sake of transparency, I'm the author of the two latter papers, and I'm not asking the authors to add them as references. My point is that there is a need for clarification, that pure pattern scaling is limited to uncertainties in modelling for probabilistic assessment, and going further with natural variability requires additional statistical tools."

We have taken out the word probabilistic here, and have a longer discussion of probabilistic extensions later in the text where we mention the works suggested here (although we were in fact already citing them in our first version of the article).

"L38-41: In STITCHES, the portions of existing simulations are found not only through the median in global mean temperature, but also by its derivative."

Thanks for pointing this out, we have now fixed this in the text.

"L56-58: The feedbacks of JULES don't feedback in FaIR or PRIME (yet)."

We have now made it clearer that such feedback is a potential extension for prime and not an existing feature.

"L17-137: At resolution of the ESM? Is there any rescaling?"

METEOR works on the resolution of the ESM model. For the comparison and plots in the manuscript we have done rescaling for multi-model comparisons, but the model natively emulates at the model resolution, and updated error score tables have also been calculated at model resolution. We added a note to clarify this to the text.

"L140-143: decomposing xglobal with IRFs is a good idea, it simplifies the modelling10. Another approach is to decompose the local effects with IRFs, e.g. TGHG, as conducted in Womack et al 2025 11 . In my opinion, both approaches have pros and cons, I'm not sure which one performs best, but Womack et al, 2025 should be mentioned."

Thank you for pointing out this very relevant reference, we have included references to this now.

"L211: the link between the Moore-Penrose pseudoinverse and Barata and Hussen, 201212 should be clearer. In the text, the paper was mentioned without the method, and here the method is mentioned without the paper."

We have tried to improve this link now by citing in both places and mentioning the technique where the citation was already in place.

---

## Referee Report (RR1)

**2nd Review for**

**METEORv1.0.1: A novel framework for emulating multitimescale regional climate responses**

The new version of this manuscript has improved. The issues that I had raised are mostly solved. The only remaining points are:

- The current justification for timescales with a lower limit of 1 year are not physically/statistically valid. I argue below why the temporal resolution does not preclude lower timescales. I recommend mentioning this as a modelling choice for this statistical model in the Discussion.
- In the Discussion, please point out to the limits in regional performance.

Regarding the local validation / performance of the emulator, the authors combined my suggestions with those of Referee 1 with new figures & sections. While there is no map of the R2, the maps of the difference between CMIP6 mean and emulation mean (Figs 8-9) and the plots that mention R2 (Figs 10-11) still convey the relevant information, albeit harder to extract the limitations. For instance, the R2 ssp534-over is lower over Australia for temperatures (Fig 10h) but it does not really appear on map (Fig 8i), and ssp534-over is not in Figure 14. In my opinion, the last two decades of ssp534-over are not representative of the difficult part of the overshoot, which is right after the peak in warming. I will not ask the authors to add new figures, there are already enough. I simply suggest to discuss a bit more the (local/)regional performance than it is now. Section 3.4 provide many info, but it is rather at a global scale. For now, it is only Line 355-356 of the version with tracked changes where we read about lower performance in some regions. Could these new results make it to the Discussion?

I fully agree that the assumption for the dependency of IRF on preindustrial/current conditions is classic and well understood. I acknowledge that increasing the complexity of the training would represent a massive additional work, and that the work presented in this manuscript is sufficient for publication. Same goes for using PDRMIP experiments.

Regarding the timescales limited to 1 year, I would not interpret that annual resolution for the inputs forces us to timescales higher than 1 year (L408-411). In short, we can infer the pace of processes even if their characteristic timescale is shorter than the pace of observations.

In more details, while it is known that there is a hysteresis, thus timescales higher than 0, we cannot assume that the physical value would be below 1. For instance, a very small hysteresis implies timescales that tend toward 0+, i.e. where the IRF tends towards a Dirac, i.e. where the convolution becomes equivalent to a pattern scaling. Very importantly, imposing a limit to 1

imposes a limit to what the IRF can do. For instance, an IRF  $y=e^{-(t-t')/2}$  cannot give less than a value of 0.37 to the forcing at t'=1, while data may say that it could be lower in some

situations. If data shows something closer to 0.14, it may be a timescale around 0.5 year. There is of course a limitation from the data, which is if the timescale is so small that the value at t'=1 becomes non significant.

The fact that so many models hit the set threshold of the timescale at 1 raises a flag, that it is possible that the term with the lowest timescale may be a quick equilibrium, e.g. in the range of 0.1-1. Imposing a spurious limit has the opposite effect, reducing the validity of the approach, while not being less interpretable.

To summarize, the explanations L408-411 are not valid from a physical and statistical perspective. I am not asking the authors to retrain the model, but I encourage them to mention that this modelling choice as a potential limitation.

Regarding all the other points, the manuscript gained in clarity.